# EXCHANGEABLE DATASET AMORTIZATION FOR BAYESIAN POSTERIOR INFERENCE

## ABSTRACT

Bayesian inference is a natural approach to reasoning about uncertainty. Unfortunately, in practice it generally requires expensive iterative methods like MCMC to approximate posterior distributions. Not only are these methods computationally expensive, they must be re-run when new observations are available, making them impractical or of limited use in many contexts. In this work, we amortize the posterior parameter inference for probabilistic models by leveraging permutation invariant, set-based network architectures which respect the inherent exchangeability of independent observations of a dataset. Such networks take a set of observations explicitly as input to predict the posterior with a single forward pass and allow the model to generalize to datasets of different cardinality and different orderings. Our experiments explore the effectiveness of this approach for both posterior estimation directly as well as model predictive performance. They show that our approach is comparable to dataset-specific procedures like Maximum Likelihood estimation and MCMC on a range of probabilistic models. Our proposed approach uses a reverse KL-based training objective which does not require the availability of ground truth parameter values during training. This allows us to train the amortization networks more generally. We compare this approach to existing forward KL-based training methods and show substantially improved generalization performance. Finally, we also compare various architectural elements, including different set-based architectures (DeepSets vs Transformers) and distributional parameterizations (Gaussian vs Normalizing Flows).

## 1 INTRODUCTION

Bayesian analysis of data has become increasingly popular and widely used in numerous scientific disciplines. In politics, predictive models based on public polling and other factors play a crucial role in the discourse around the state of a campaign. Throughout the COVID-19 pandemic, models that estimate the infectiousness of the virus, the efficacy of public health measures, and the future course of the pandemic became critical to government planning and the public's understanding of the pandemic. In cryogenic electron microscopy (cryo-EM), the posterior over an unknown 3D atomic-resolution molecular structure is explored given the 2D image observations.

While recent years have brought improved software which has made this analysis more accessible to statistical practitioners (Bingham et al., 2019; Carpenter et al., 2017; Štrumbelj et al., 2023), the analyses still remain computationally burdensome. Further, in practical contexts where new observations are continuously available, the analysis must be re-run every time new data becomes available, e.g., when new case counts become available, previous measurements are corrected, or when applied to different geographic regions. As a result practitioners adopt approximations (Welling & Teh, 2011; Gelfand, 2000; Brooks, 1998), simplify their models (Hoffman et al., 2013; Blei et al., 2017) or reduce the frequency with which they perform their analyses.

A common thread is that the probabilistic model defining the relationship between the unknown parameters and the observed data is fixed. Poll aggregation models use hierarchical time series models, infectious diseases are studied using variations on compartment models, and cryo-EM uses a linear image formation model. This makes these applications ideal candidates for amortized inference (Morris, 2013; Paige & Wood, 2016; Kingma & Welling, 2013; Rezende et al., 2014; Stuhlmüller et al., 2013). In this paper, we address the problem of Bayesian posterior estimation

(a) Mean of Gaussian  (b) Linear Regression  (c) Nonlinear Regression  (d) Gaussian Mixture  (e) Linear Classification  (f) Nonlinear Classification

Figure 1: **Fixed-Dimension Visualization:** Illustration of proposed approach when trained on a fixed dimensional observation space. Samples from the inference model are shown in red, data points in blue, and ground truth in black. For classification, data points are colored by their ground-truth labels, and decision boundary corresponds to an ensemble of samples from the inference model. We see that the learned amortized variational distribution appropriately captures the underlying distributions.

through the use of amortized inference, which will allow for efficient and principled methods for posterior analysis, which can consequently be utilized for modeling predictions.

To do this we propose using neural networks to learn a function that maps an observed *dataset* directly to the corresponding posterior distribution, without the need to perform explicit Bayesian inference, e.g., with Markov chain Monte Carlo (MCMC) sampling (Gelfand, 2000; Hoffman et al., 2014). This mapping, if learned properly, allows generalization to different datasets for the same underlying model. It also has the potential to generalize further to a range of domains since many models are widely standard, e.g., regression and classification models. Motivated by work in set-based neural network architectures like Transformers and DeepSets (Zaheer et al., 2017; Vaswani et al., 2017; Lee et al., 2019), we design a function that can take an arbitrary set as input as opposed to an ordered list of fixed length and explore a number of different choices which respect permutation invariance with respect to the ordering of observations when modeling the Bayesian posterior.

Our primary motivation is posterior inference as the parametric values themselves are often of interest in applied statistical practice, e.g., for assessing the success of a pandemic intervention or the impact of a factor on public opinion polling. Additionally, we demonstrate the utility of our proposed approach in a closely related problem of posterior prediction where the goal is to model future predictions given some observations. In practice real-world datasets do not exactly follow standard models, e.g., while practitioners are interested in the results of linear regression models, data rarely follows such models exactly. As a result, previous amortization approaches (Radev et al., 2020) which rely on training with datasets and their corresponding known parameters may struggle to generalize to data of practical use. Instead, we propose a new training objective that can operate solely on datasets without knowing the corresponding parameters, thereby allowing for a wider diversity of data to be incorporated during training for better generalization to real-world settings. Through our experiments, we establish the superiority of our proposed approach. Our contributions include

- Proposing a novel method for performing explicit Bayesian posterior estimation in known probabilistic models solely through inference on an amortization network trained via the reverse KL framework, and demonstrating its effectiveness in a variety of settings and with several well-known probabilistic models.
- Providing insights into various design choices like the architectural backbone used and the choice of parametric distribution through detailed ablation experiments.
- Highlighting the superior performance of our proposed approach when compared to existing baselines, especially in the presence of model misspecification and real-world data.

## 2 BACKGROUND

Formally, we consider the problem of estimating the posterior distribution $p(\boldsymbol{\theta}|\mathcal{D})$ for known probabilistic models $p(\boldsymbol{x}, \boldsymbol{\theta})$, where $\boldsymbol{x} \in \mathbb{R}^d$ is observed through $n$ independent and identically distributed (*iid*) samples $\mathcal{D} = \{\boldsymbol{x}_i\}_{i=1}^n$ and $\boldsymbol{\theta} \in \mathbb{R}^k$ denotes the parameters of the model.

For a given probabilistic model $p(\boldsymbol{x}, \boldsymbol{\theta})$ and observed *iid* samples $\mathcal{D}$, Bayesian inference refers to the problem of estimating the posterior distribution $p(\boldsymbol{\theta}|\mathcal{D})$. This estimation problem boils down to an application of Bayes' rule

$$p(\boldsymbol{\theta}|\mathcal{D}) = \frac{p(\boldsymbol{\theta})}{p(\mathcal{D})} \prod_{i=1}^n p(\boldsymbol{x}_i|\boldsymbol{\theta}). \tag{1}$$

| | | Model | $L_2$ Loss ($\downarrow$) | | | | | | | Accuracy ($\uparrow$) | | | |
| | $q_\varphi$ | | GM | | GMM | LR | | NLR | | LC | | NLC | |
| | | | 2D | 100D | 5D 2 cl | 1D | 100D | 1D | 25D | 2D | 100D | 2D | 25D |
|---|---|---|---|---|---|---|---|---|---|---|---|---|---|
| Baseline | - | Random | 5.829 | 301.4 | 5.14 | 2.844 | 69.4 | 42.6 | 289 | 51.4 | 50.0 | 49.8 | 49.8 |
| | - | Optimization | 1.989 | 101.2 | 0.42 | 0.256 | 8.62 | 0.29 | 30.5 | 94.0 | 70.3 | 96.5 | 79.0 |
| | - | MCMC | 2.055 | 106.3 | 0.58 | 0.282 | 11.5 | N/A | 39.5 | 92.6 | 63.5 | 95.8 | 72.0 |
| Fwd-KL | Gaussian | DeepSets | 2.014 | 103.5 | 2.44 | 0.263 | 52.2 | 31.2 | 243 | 80.5 | 49.8 | 59.2 | 57.7 |
| | | Transformer | 2.013 | 103.2 | 2.46 | 0.263 | 20.6 | 32.0 | 233 | 80.1 | 62.2 | 60.0 | 57.5 |
| Rev-KL | | DeepSets | 2.012 | 102.8 | 0.47 | 0.262 | 25.9 | 0.34 | 43.3 | 92.6 | 58.8 | 90.5 | 60.4 |
| | | Transformer | 2.013 | 102.5 | 0.46 | 0.264 | 11.8 | 0.35 | 31.4 | 92.5 | 65.7 | 90.3 | 74.8 |
| Fwd-KL | Flow | DeepSets | 2.015 | 103.6 | 0.60 | 0.263 | 52.6 | 11.4 | 186 | 93.6 | 50.0 | 61.5 | 58.5 |
| | | Transformer | 2.013 | 103.2 | 0.67 | 0.264 | 21.2 | 12.6 | 182 | 93.7 | 63.7 | 69.5 | 58.4 |
| Rev-KL | | DeepSets | 2.011 | 102.7 | 0.50 | 0.262 | 29.2 | 0.33 | 74.9 | 93.3 | 56.6 | 90.8 | 61.0 |
| | | Transformer | 2.017 | 102.7 | 0.52 | 0.263 | 11.2 | 0.33 | 31.2 | 93.1 | 66.2 | 91.5 | 75.1 |

Table 1: **Fixed-Dimension Posterior Prediction:** Experimental results for posterior inference on fixed dimensional datasets evaluated on estimating the (a) mean of a Gaussian (GM), (b) means of Gaussian mixture model (GMM), (c) parameters for (non-)linear regression (NLR/LR), and (d) parameters for (non-)linear classification (NLC/LC). We consider different backbone architectures and parametric distributions $q_\varphi$, and use dataset-specific Bayesian and point estimates as baselines. $L_2$ Loss and Accuracy refer to the expected posterior-predictive $L_2$ loss and accuracy respectively.

Analytically computing Equation 1 is problematic since the normalization constant requires computing the marginal $p(\mathcal{D}) = \int_{\boldsymbol{\theta}} p(\boldsymbol{\theta}, \mathcal{D}) \, d\boldsymbol{\theta}$, which is often intractable. Thus, practitioners rely on approximate approaches to estimating the posterior, namely sampling and variational inference (VI). Sampling estimates the posterior through finite points which can be obtained by constructing a Markov chain whose asymptotic stationary distribution is the true posterior. In this work, we instead focus on amortizing the computations done in VI, which approximates the posterior by searching for the closest distribution within a parametric family of distributions.

VI methods approximate the true posterior $p(\boldsymbol{\theta}|\mathcal{D})$ with a variational distribution $q_\varphi(\boldsymbol{\theta})$ and convert the estimation problem into the following optimization problem

$$\varphi^* = \arg\min_{\varphi} \mathbb{KL}[q_\varphi(\cdot)||p(\cdot|\mathcal{D})] \tag{2}$$

which is equivalent to optimizing the Evidence Lower-Bound (ELBO)

$$\varphi^* = \arg\max_{\varphi} \mathbb{E}_{\boldsymbol{\theta} \sim q_\varphi(\cdot)} \left[ \log \frac{p(\mathcal{D}, \boldsymbol{\theta})}{q_\varphi(\boldsymbol{\theta})} \right] \tag{3}$$

For any given probabilistic model, computing the Bayesian posterior for a new dataset requires solving a new, often iterative, optimization problem to learn $q_\varphi(\cdot)$, which is implicitly a function of $\mathcal{D}$. This limits the approach computationally and makes knowledge transfer among different tasks or datasets infeasible due to independent optimization problems.

Variational Autoencoders (VAEs) (Kingma & Welling, 2013; Rezende et al., 2014; Rezende & Mohamed, 2015) bypass this problem in latent-variable models by amortizing the variational distribution explicitly on different data points. That is, given a probabilistic model

$$p(\boldsymbol{x}, \boldsymbol{z}) = p(\boldsymbol{x}|\boldsymbol{z}) \, p(\boldsymbol{z}) \tag{4}$$

with $\boldsymbol{z}$ as the latent variable, they consider $q_\varphi(\boldsymbol{z}|\boldsymbol{x}_i)$ as the variational distribution, where the conditioning is explicitly done on $\boldsymbol{x}_i$ by predicting the parameters of the variational distribution from $\boldsymbol{x}_i$, e.g., $\mathcal{N}(\cdot|\boldsymbol{\mu}_\varphi(\boldsymbol{x}_i), \boldsymbol{\Sigma}_\varphi(\boldsymbol{x}_i))$. Essentially, this contrasts considering the variational distribution as $q_{\varphi(\boldsymbol{x}_i)}(\boldsymbol{z})$ instead of $q_{\varphi_i}(\boldsymbol{z})$ for the observation $\boldsymbol{x}_i$, where the former is more scalable since the approximation for a new observation $\boldsymbol{x}_j$ is obtained directly from inference over the network $\varphi(\cdot)$ as opposed to solving a new optimization problem, which is done in the latter. Such models are largely successful owing to the generalization capabilities of neural networks to new unseen observations as long as the encoder $q_\varphi(\boldsymbol{z}|\boldsymbol{x}_i)$ is trained on enough diverse observations $\boldsymbol{x}'_i$s.

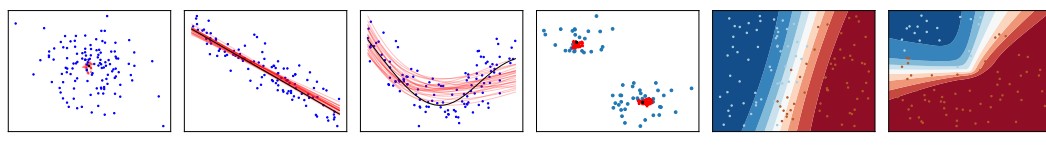

| (a) Mean of Gaussian | (b) Linear Regression | (c) Nonlinear Regression | (d) Gaussian Mixture | (e) Linear Classification | (f) Nonlinear Classification |

Figure 2: **Variable-Dimension Visualization:** Illustration of proposed approach when trained on a variable observational space. Samples from the inference model are shown in red, data points in blue, and ground truth in black. For classification, data points are colored by their ground-truth labels, and decision boundary corresponds to an ensemble of samples from the inference model. We see that the learned amortized variational distribution appropriately captures the underlying distributions.

Taking inspiration from VAEs and their use of amortization, we return to the more general problem of learning Bayesian posteriors for probabilistic models through VI. In contrast to the typical VAE setup where the probabilistic model is also learned, we consider a known probabilistic model and rely on amortization at the dataset level $\mathcal{D}$ instead of single data points $\boldsymbol{x}_i$ to directly obtain the approximate Bayesian posterior. Since the encoder $q_\varphi$ takes a set of observations $\mathcal{D}$ as input, it has to satisfy the exchangeability criteria implicit in $\mathcal{D}$ as the observations are *iid*. Recent advances in permutation invariant, set-based architectures (Zaheer et al., 2017; Vaswani et al., 2017) can be leveraged in efficiently designing such an encoder.

Prior work trains the encoder by either minizming the forward KL, $\mathbb{KL}\left[p(\cdot|\mathcal{D})||q_\varphi(\cdot|\mathcal{D})\right]$ (Radev et al., 2020) or performing Bayesian inference on some latent variables in predictive systems (Garnelo et al., 2018b). The former cannot handle training with data whose underlying model is unknown and hence cannot deal with model misspecification but enjoys the benefits of not requiring a computable likelihood. The latter is predominantly designed for predictive modeling and thus cannot be used to provide useful information and uncertainty about model parameters. We propose a fully Bayesian approach, which, like (Garnelo et al., 2018b) requires a computable and differentiable likelihood and reparameterizable $q_\varphi$ but approximates the posterior through an explicit form in the parameter space and can be used in cases of model misspecification. We refer the readers to Appendix A for a more comprehensive discussion about related work.

## 3 METHOD

Our goal is to train a system that approximates the posterior distribution $p(\boldsymbol{\theta}|\mathcal{D})$ given a dataset $\mathcal{D} := \{\boldsymbol{x}_1, \boldsymbol{x}_2, \cdots, \boldsymbol{x}_n\} \subseteq \mathbb{R}^d$ where $\boldsymbol{x}_i \sim p(\boldsymbol{x}|\boldsymbol{\theta})$. We achieve this by learning an amortized variational distribution $q_\varphi(\boldsymbol{\theta}|\mathcal{D})$ conditioned explicitly on the full datasets. Similar to standard VI approaches, we can train $q_\varphi$ by minimizing the $\mathbb{KL}$ divergence between the approximate and the true posterior, i.e., $\mathbb{KL}[q_\varphi(\cdot|\mathcal{D})||p(\cdot||\mathcal{D})]$ which reduces to maximizing the ELBO:

$$\arg \max_\varphi \mathbb{E}_{\boldsymbol{\theta} \sim q_\varphi(\cdot|\mathcal{D})} \left[\log \frac{p(\mathcal{D}, \boldsymbol{\theta})}{q_\varphi(\boldsymbol{\theta}|\mathcal{D})}\right] \tag{5}$$

While this is the case for VI on a single dataset, we are interested in generalizing to a collection of datasets $\{\mathcal{D}_i\}_{i=1}^K$, or even more generally to a family of datasets obtained from some dataset generating distribution $\chi$ such that $\mathcal{D} \sim \chi$. Much like VAEs learn to amortize over data points by training with ensembles of such points, our hope is to amortize over data sets by training with $\{\mathcal{D}_i\}_{i=1}^K$. To obtain the posterior distribution for each dataset, we consider a mean-field assumption over the variational distribution $q_\varphi$ and an iid assumption over the datasets. This factorizes the variational distribution $q_\varphi(\boldsymbol{\theta}_1, ...\boldsymbol{\theta}_K|\mathcal{D}_1, ..., \mathcal{D}_K)$ into $\prod_{i=1}^K q_\varphi(\boldsymbol{\theta}_i|\mathcal{D}_i)$ and the likelihood model as $p(\mathcal{D}_1, ..., \mathcal{D}_K|\boldsymbol{\theta}_1, ..., \boldsymbol{\theta}_K) = \prod_{i=1}^K p(\mathcal{D}_i|\boldsymbol{\theta}_i)$. The ELBO in this setting can be written as:

$$\arg \max_\varphi \mathbb{E}_{\boldsymbol{\theta}_1, ..., \boldsymbol{\theta}_K \sim \prod_{i=1}^K q_\varphi(\cdot|\mathcal{D}_i)} \left[\log \prod_{i=1}^K \frac{p(\mathcal{D}_i, \boldsymbol{\theta}_i)}{q_\varphi(\boldsymbol{\theta}_i|\mathcal{D})}\right] \tag{6}$$

In particular, given a dataset generating distribution $\chi$, we can re-write the objective in Equation 6 as:

$$\tilde{\varphi} := \arg \max_\varphi \mathbb{E}_{\mathcal{D} \sim \chi} \mathbb{E}_{\theta \sim q_\varphi(\cdot|\mathcal{D})} \left[\log \frac{p(\mathcal{D}, \boldsymbol{\theta})}{q_\varphi(\boldsymbol{\theta}|\mathcal{D})}\right] . \tag{7}$$

This naturally introduces a dependency on the dataset generating distribution $\chi$. Since we are working with a known probabilistic model, an obvious choice of $\chi$ is $p(n)p(\boldsymbol{\theta}) \prod_{i=1}^n p(\boldsymbol{x}_i|\boldsymbol{\theta})$, samples from

| | | | L₂ Loss (↓) | | | | | | | Accuracy (↑) | | | |
|---|---|---|---|---|---|---|---|---|---|---|---|---|---|
| | $q_\varphi$ | Model | GM | | GMM | LR | | NLR | | LC | | NLC | |
| | | | 2D | 100D | 5D 2 cl | 1D | 100D | 1D | 50D | 2D | 100D | 2D | 50D |
| Baseline | - | Random | 6.255 | 299.4 | 4.62 | 2.908 | 69.4 | 41.3 | 550 | 50.0 | 49.9 | 50.9 | 49.8 |
| | - | Optimization | 2.020 | 100.8 | 0.42 | 0.258 | 8.00 | 0.31 | 96.2 | 92.6 | 71.6 | 96.8 | 74.2 |
| | - | MCMC | 2.214 | 109.4 | 0.84 | 0.369 | 12.3 | N/A | 104 | 89.7 | 62.5 | 96.5 | 67.1 |
| Fwd-KL | Gaussian | DeepSets | 2.222 | 142.7 | 2.37 | 0.269 | 55.6 | 29.8 | 464 | 80.3 | 50.5 | 59.3 | 59.3 |
| | | Transformer | 2.387 | 110.7 | 2.41 | 0.276 | 23.3 | 29.9 | 453 | 77.9 | 62.2 | 60.3 | 59.3 |
| Rev-KL | | DeepSets | 2.049 | 105.2 | 0.47 | 0.270 | 24.5 | 0.57 | 149 | 90.2 | 58.6 | 89.6 | 62.7 |
| | | Transformer | 2.060 | 104.8 | 0.46 | 0.267 | 11.9 | 0.55 | 83.5 | 90.6 | 66.0 | 89.3 | 72.5 |
| Fwd-KL | Flow | DeepSets | 2.486 | 140.0 | 0.59 | 0.269 | 59.2 | 24.0 | 379 | 91.1 | 50.2 | 60.2 | 60.5 |
| | | Transformer | 2.297 | 110.2 | 0.46 | 0.272 | 24.9 | 20.8 | 367 | 86.7 | 63.2 | 60.9 | 60.8 |
| Rev-KL | | DeepSets | 2.049 | 109.8 | 0.51 | 0.269 | 29.0 | 0.57 | 150 | 91.2 | 56.2 | 88.2 | 63.5 |
| | | Transformer | 2.049 | 105.1 | 0.45 | 0.267 | 12.7 | 0.47 | 87.6 | 91.1 | 67.0 | 90.0 | 71.7 |

Table 2: **Variable-Dimension Posterior Prediction:** Experimental results for posterior inference on variable dimensional datasets evaluated on estimating the (a) mean of a Gaussian (GM), (b) means of Gaussian mixture model (GMM), (c) parameters for (non-)linear regression (NLR/LR), and (d) parameters for (non-)linear classification (NLC/LC). We consider different backbone architectures and parametric distributions $q_\varphi$, and use dataset-specific Bayesian and point estimates as baselines. $L_2$ Loss and Accuracy refer to the expected posterior-predictive $L_2$ loss and accuracy respectively.

which can be obtained using ancestral sampling. Here, $n$ is the dataset cardinality and $p(n)$ is a distribution over positive integers. Thus, given any probabilistic model, obtaining a dataset generating distribution is easy and just relies on sampling from the probabilistic model itself. However, $\chi$ can also be obtained from other sources, for example, a stream of real-world data, through interventions on the data-generating process, or through bagging on some large real-world dataset.

An equivalent way of looking at the optimization problem described in Equation 7 is minimizing the Kullback–Leibler (KL) divergence between the approximate and the true posterior

$$\tilde{\varphi} = \arg\min_\varphi \mathbb{E}_{\mathcal{D}\sim\chi} \mathbb{KL}\left[q_\varphi(\cdot|\mathcal{D})||p(\cdot|\mathcal{D})\right] \qquad (8)$$

which leads to a *Reverse KL* optimization objective[1]. An alternative approach to approximating this amortized posterior can be obtained by minimizing the *Forward KL* objective (Radev et al., 2020) which we consider as a baseline comparison. It requires access to the ground truth parameters $\boldsymbol{\theta}$ and leads to a different objective owing to the asymmetric nature of the divergence.

$$\hat{\varphi} = \arg\min_\varphi \mathbb{E}_{\mathcal{D}\sim\chi} \mathbb{KL}\left[p(\cdot|\mathcal{D})||q_\varphi(\cdot|\mathcal{D})\right] \qquad (9)$$

Since VI solutions search for an optimal distribution within a parametric family of distributions, the choice of the family of distributions specified by $q_\varphi$ is also dependent on the user and can be partially specified based on the problem setup. For our experiments, we either model $q_\varphi$ as a Gaussian distribution, which requires learning the mean $\boldsymbol{\mu}_\varphi : \mathcal{D} \to \mathbb{R}^m$ and covariance matrix $\boldsymbol{\Sigma}_\varphi : \mathcal{D} \to \mathbb{R}^{m \times m}$ functions, or as a normalizing flow conditioned on the dataset $\mathcal{D}$.

Having defined all the pieces necessary for amortized inference, we now look at architectures that operate on sets of observations. Importantly, the architectures should respect the exchangeability of the points in a dataset, as the observations are iid, and be generalizable to different dataset cardinality. That is, such an architecture should map different orderings of the observations to the same posterior distribution. We rely on two architectures that satisfy this permutation invariance: DeepSets and Transformers, and defer details regarding their ability of respecting exchangeability to Appendix B.

## 4 EXPERIMENTS

Our approach relies on the user specifying a probabilistic model which describes their belief about the data-generating process as well as the parameters of interest. To showcase the wide applicability of our

---

[1]Given the equivalence of Equations 7 and 8, we will use amortized VI and Reverse KL interchangeably.

| $q_\varphi$ | Data | Linear | MLP Nonlinear | GP Nonlinear |
|---|---|---|---|---|
| | Model | NLR | LR | NLR |
| Fwd-KL | Gaussian | 14.369 | 3.791 | 13.906 |
| Rev-KL | | 0.340 | 2.354 | 0.128 |
| *+ trained on switched data* | | 0.330 | 0.637 | 0.099 |
| Fwd-KL | Flow | 12.514 | 3.308 | 12.385 |
| Rev-KL | | 0.323 | 1.722 | 0.087 |
| *+ trained on switched data* | | 0.321 | 0.641 | 0.080 |

Table 3: **Model Misspecification:** Posterior predictive performance with $L_2$ loss metric on OoD evaluation. Data indicates the data-generating function used to evaluate the probabilistic models (Model), which were trained on their corresponding data-generating functions. We see that reverse KL is better able to handle OoD data, and can also leverage it for training (*switched*).

approach, we perform experimentation on different well-known probabilistic models encompassing both supervised and unsupervised scenarios. In particular, we look at the following problems of estimating the Bayesian posterior over the (a) mean of a Gaussian distribution (GM), (b) means of a Gaussian mixture model (GMM), (c) parameters of a (non-)linear regression model (NLR/LR), and (d) parameters of a (non-)linear classification model (NLC/LC). We refer the readers to Appendix C for more details about the particulars of each probabilistic model, including their closed form likelihoods and the priors considered. Throughout our experiments, we observe superior performance of our proposed approach, especially in problems with high-dimensional and multi-modal posteriors.

In all our experiments, we generally consider two types of baselines: dataset-specific and amortized. For dataset-specific baselines, we perform maximum likelihood estimation using gradient-based optimization as well as an approximate Bayesian inference procedure through Langevin-based MCMC sampling, which also uses the gradient information. Such baselines rely on iterative procedures and must be run independently for different datasets. On the other hand, we also consider a forward KL based amortized baseline, which relies on training a similar inference model $q_\varphi$ but with the forward KL optimization criteria (Equation 9) as well as a random baseline, which provides an estimate of performance under the prior distribution. We refer the readers to Appendices F, D and H for details about the experiments, metrics and additional results respectively.

## 4.1 FIXED-DIMENSION EXPERIMENTS

Given a known probabilistic model $p(\mathcal{D}, \boldsymbol{\theta})$, we train an amortized inference model $q_\varphi(\cdot|\mathcal{D})$ to approximate the often intractable posterior over the parameter $\boldsymbol{\theta}$ for different datasets with varying cardinality. It is, however, important to note that the probabilistic model implicitly describes the dimensionality of the observations and thus requires training a different model for observations lying in different dimensional spaces. For example, an inference model trained to predict the posterior over linear regression weight vectors for 1-dimensional observations cannot be naively used to also obtain the Bayesian posterior for 2-dimensional observations.

We train separate amortized variational inference models for the different probabilistic model classes and number of feature dimensionalities. The zero-shot posterior predictive performance of our proposed approach for low-dimensional data is visualized in Figure 1. Our results highlight that in low-dimensional setups, the amortized inference model captures the underlying trends in the data reasonably well for all of the probabilistic models that we consider.

Next, we empirically evaluate the performance of such systems in more complex, high-dimensional setups for the same set of probabilistic models. For each task, we evaluate all the methods on 100 test datasets sampled in-distribution from the data generating distribution $\chi$. We compare the posterior predictive performance of the proposed amortized variational inference systems (based on either $L_2$ loss or accuracy metric) to three dataset-specific baselines: Random, Optimization, and MCMC. Contrary to dataset-specific routines, we also consider an amortized inference baseline trained under the forward KL paradigm and compare it to our proposed approach, reverse KL. We consider modeling $q_\varphi$ either as diagonal Gaussian or normalizing flows for both the amortization setups. Table 1 shows that the reverse KL amortization outperforms the forward KL one and is often quite comparable to dataset-specific baselines. We refer the readers to Appendix F.1 and H.1 for experimental details and additional results highlighting the superiority of our proposed method.

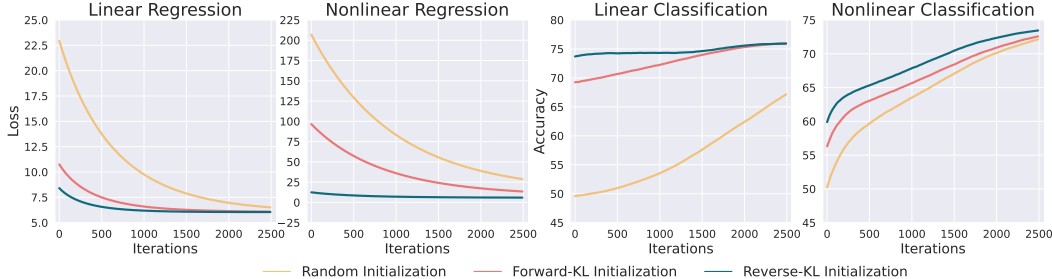

Figure 3: **Tabular Experiments:** Initializing parameters from the amortized model, especially Reverse KL, leads to good zero-shot performance across (non-)linear regression and classification. These benefits persist over the course of optimization.

## 4.2 VARIABLE-DIMENSION EXPERIMENTS

To alleviate the problem of having to train different models for data of different dimensionalities, we consider training a single amortized inference model by leveraging masking of redundant dimensions to handle datasets with variable number of features. For example, with a maximum dimensionality of 100, a dataset with five dimensions can be visualized as being embedded in a 100-dimensional space with 95 dimensions constantly being zero. This is useful because it not only allows us to train a single model capable of, for example, handling linear regression problems of arbitrary feature dimensions but also allows the amortization system to leverage the underlying commonality in the solution for such tasks (e.g., the structure of the solution for linear regression remains the same, i.e., $(\boldsymbol{X}^T \boldsymbol{X})^{-1} \boldsymbol{X}^T \boldsymbol{y}$, irrespective of the number of features in $\boldsymbol{X}$).

We train the same amortized inference models as done for fixed-dimensional observations but now on observations with a maximum of 100 feature dimensions. Samples from this approximate posterior for low-dimensional settings are visualized in Figure 2, showing that the amortized model provides reasonable estimates. We extend our analysis to high-dimensional setups and validate the performance over 100 test datasets sampled from $\chi$, which is provided in Table 6. Like before, we consider both dataset-specific and forward KL amortization models as baselines, with both diagonal Gaussian and normalizing flows-based parameterization for the approximate distribution. Details about the experimental setup and additional results are provided in Appendix F.2 and H.2. We emphasize that in high-dimensional and multi-modal settings (eg. NLR/NLC), our approach is quite superior to forward KL based approach, and often comparable to dataset-specific baselines.

## 4.3 MODEL MISSPECIFICATION

As noted in prior work (Müller et al., 2021; Hollmann et al., 2022), the performance and use-case of such systems, when trained with simulated data, relies heavily on the choice of prior in describing the dataset generating function $\chi$. In particular, if the space of functions seen during training differs greatly from those seen in real-world setups, the amortized inference model would struggle to provide a reliable posterior for the assumed probabilistic model. This becomes an increasingly significant problem as the observation and hypothesis space scale up due to the curse of dimensionality. Thus, it becomes essential to test such methods in out-of-distribution (OoD) settings where the ground-truth probabilistic model might differ from the one assumed in modeling the data. We look at the effect of such distribution shifts on our approach, as well as provide a detailed analysis of how such shifts can be incorporated into the training paradigm under our proposed variational setup but cannot be under the existing forward KL based approaches and its resulting implications.

Thus, we now shift our focus to the scenario where we train the amortization system with the dataset generating distribution $\chi$ obtained from the assumed probabilistic model but the actual data seen during evaluation $\chi'$ stems from a different underlying model. For example, we can obtain the posterior from an amortization model assuming an underlying linear system, but our evaluation data might instead come from a nonlinear system. Our goal is to evaluate if the inference model still gives a reliable estimate of the posterior as the computations required for Bayesian inference remain the same even if the data is nonlinear (eg. the analytical term for the Bayesian posterior for linear regression is invariant to the specifics of the data, i.e. whether it is linear or not). Even further, we often have large amounts of data without access to its ground-truth generating process. For example, we might assume the ground-truth process to be a 1-layer MLP with some noise, but in reality, the data might

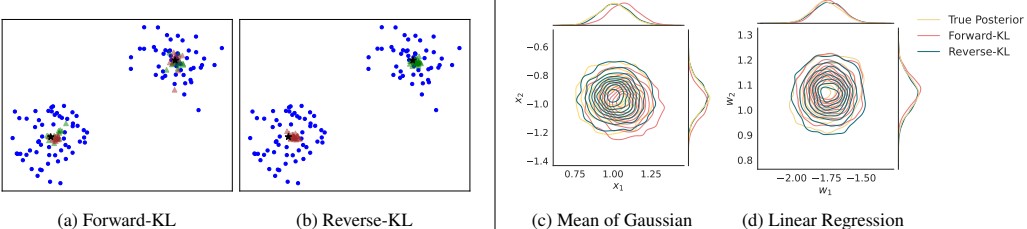

(a) Forward-KL       (b) Reverse-KL       (c) Mean of Gaussian       (d) Linear Regression

Figure 4: **Left**: Estimation of the means of a GMM, where red and green samples denote the first and second mean vectors. Unlike in reverse KL, the cluster labels switch in forward KL, highlighting its ability to capture underlying multi-modality. **Right**: Kernel density estimation of the true posterior, overlaid with estimates from forward and reverse KL systems, for different probabilistic models.

come from a Gaussian Process (GP). Such cases of misspecification are often observed when the data comes from underlying physics or biology-based processes, or through experimentation, where knowing the exact underlying generative process is an exceptionally hard problem, if not impossible.

In such cases, training with the forward KL setup is not realizable as it requires access to paired data in the form $(\mathcal{D}, \boldsymbol{\theta})$. At best, such a model can only be trained with data simulated from the assumed probabilistic model $\chi$, which is sub-optimal as the unpaired data $\chi'$ provides meaningful information about the underlying process but has to be discarded. Additionally, the simulated data may be quite different and the resulting amortized model might not generalize to the actual data that we care about. In contrast, we can leverage our proposed reverse KL approach to train an amortized inference model to predict the posterior over the assumed probabilistic model's parameters by directly using the available unpaired data $\chi'$ during training. We highlight in Table 3 that forward KL based approaches struggle more than our proposed approach on zero-shot evaluation in OoD setups. Further, when our proposed approach is trained directly on the available data, it leads to even better performance, even when the assumed probabilistic model is incorrect. For example, we see that when we try to model nonlinear data using a linear model, reverse KL leads to better performance which drastically improves if we train directly on the nonlinear data, while considering a linear model throughout. We refer the readers to Appendix F.3 and H.3 for experimental details and additional results on model misspecification.

## 4.4 TABULAR EXPERIMENTS

While we have highlighted the benefits of our approach of amortizing Bayesian inference over a wide variety of probabilistic models, it is still an important question to consider its relevance to real-world scenarios. To answer this question, we consider a suite of tasks from the OpenML platform for both regression and binary classification. In particular, we filter out tasks from the *OpenML-CTR23 - A curated tabular regression benchmarking suite* (Fischer et al., 2023) for the regression and *OpenML-CC18 Curated Classification benchmark* (Bischl et al., 2019) for the classification problems (details in Appendix G). We end up with 9 regression and 13 classification datasets with varying number of features and use the amortized inference systems trained in Section 4.2 to predict the parameters of interest conditioned explicitly on the training dataset. This can be seen as a setup for extreme domain shift / OoD setup since the amortized inference models do not see such data during training.

After initializing from the inference model, we further train the parameters of the respective probabilistic models with maximum a-posteriori and compare its performance with a corresponding model initialized from the prior and trained via Adam optimizer independently on each dataset. Through this experiment, we want to see if the amortized model provides good initializations for different real-world tasks zero-shot after training, and whether it leads to better zero-shot performance than the forward KL setup. Figure 3 shows that indeed initializing with either the forward or reverse KL model outperforms optimization from scratch, with the reverse KL formulation providing better initialization, as well as faster convergence, than the forward one for both linear and nonlinear setups. While Figure 3 only provides a normalized aggregated performance over all datasets considered, with only a diagonal gaussian $q_\varphi$, we refer the readers to Appendix H.4 for results on individual datasets with different $q_\varphi$, as well as Appendix F.4 for implementation details.

## 4.5 RESULTS

**Posterior Inference**. Making a fair comparison with the true posterior distribution is hard, as for most non-trivial problems, the posterior is intractable. However, a tractable posterior is available for the problem of estimating the mean of a Gaussian distribution as well as for Bayesian Linear Regression.

| | $q_\varphi$ | Model | Symmetric KL Divergence ($\downarrow$) | | | |
|---|---|---|---|---|---|---|
| | | | **Gaussian Mean** | | **LR** | |
| | | | *2D* | *100D* | *1D* | *100D* |
| Baseline | - | Random | 44.32 | 2339 | 244.3 | 64.1 |
| Fwd-KL | Gaussian | DeepSets | 0.024 | 0.293 | 0.061 | 36.95 |
| | | Transformer | 0.066 | 0.133 | 0.030 | 10.11 |
| Rev-KL | | DeepSets | 0.028 | 0.325 | 0.051 | 37.70 |
| | | Transformer | 0.085 | 0.082 | 0.049 | 10.66 |

Table 4: **Symmetric KL Divergence for Posterior Evaluation:** We see that for estimating the mean of a Gaussian distribution as well as for estimating the weight vector for linear regression, the amortized models are largely able to approximate the posterior well, as highlighted by the normalized symmetric KL divergence, especially when compared with the prior (Random).

We plot the kernel density estimate of the samples obtained from the true posterior, the amortized forward and the reverse KL model in Figure 4 (**Right**), which shows that both the amortization setups efficiently capture the true posterior distribution. Further, we highlight in Table 4 that, indeed, for these setups, the amortized model obtains a good approximation of the posterior distribution, however, it does worsen with increasing dimensionality. Additionally, we see from posterior predictive results that forward KL performs comparable to random chance (prior) in a lot of setups where the Bayesian posterior distribution is high-dimensional and multi-modal (Appendix H.1). In contrast, reverse KL provides reasonable estimates, alluding to its ability to better capture at least a mode of the posterior.

**DeepSets vs Transformers**. Our experiments consistently show that using Transformers as the permutation-invariant backbone architecture outperforms DeepSets, potentially because Transformers can aggregate information in a more flexible fashion than DeepSets which have a context-unaware aggregation function, e.g., sum or mean based pooling. However, we do see that in some rare cases of OoD generalization, DeepSets can outperform Transformers (Appendix H.3). To have a fair comparison, we control for the number of parameters throughout our experiments. We leave investigations into different aggregation functions (Volpp et al., 2020) as future work.

**Forward vs Reverse KL**. In our experiments on a GMM posterior, which has multiple modes because of the exchangeability of cluster labels, we see that the forward KL objective does lead to learning of a multi-modal distribution. At the same time, reverse KL only captures one mode (Figure 4, **Left**). However, in high-dimensional multi-modal settings like learning the parameters of a BNN, the forward KL objective does not lead to learning of a reasonable distribution as it attempts to cover all the modes. In contrast, the reverse KL objective does not cover multiple modes but can better model an individual mode (Tables 1 and 6; Appendix H.1 and H.2). Furthermore, unlike forward KL, the reverse KL paradigm can be trained without observing $\boldsymbol{\theta}$ but does require a computable and differentiable likelihood of the underlying, assumed model. This is useful when we don't have access to the ground-truth data-generating process but only to samples from it, as outlined in Section 4.3.

**Normalizing Flow vs Gaussian**. Our results suggest that increasing the capacity of $q_\varphi$ with normalizing flows only helps marginally for reverse KL objective. However, there is substantial improvement when incorporated in the forward KL setup. We hypothesize that given the mode-seeking tendency of reverse KL, even with the capacity to model different modes, the algorithm seeks and latches to only a single mode and capturing multiple modes in this setup is challenging.

## 5 CONCLUSION

We show that it is possible to amortize full Bayesian posterior inference for a broad class of probabilistic models and explore a variety of design decisions. In particular we show that reverse KL is effective for learning the amortization network and has significant benefits in the presence of model misspecification and in generalization to out-of-domain real-world tabular setups. We believe that this approach could provide fast insights into Bayesian posteriors in practice and help accelerate further refinements of the posterior estimates. It is an exciting direction of research that could lead to reducing the load of real-world, complex, and iterative Bayesian inference problems through quick and cheap inference over a trained amortization network. Scaling this approach to work on more complex probabilistic models is a significant focus of future work.

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

# APPENDIX

## A    RELATED WORK

In this section, we draw parallels of our work to various approaches that have been proposed to tackle the problem of either providing a good initialization for different tasks, performing implicit optimization to model predictive distributions for new tasks, or estimating the posterior through a different objective.

### A.1    VARIATIONAL AUTOENCODERS

VAEs (Kingma & Welling, 2013; Rezende et al., 2014; Rezende & Mohamed, 2015; Kingma et al., 2019) are latent variable models which model observations $\boldsymbol{x}$ conditioned on latent variables $\boldsymbol{z}$ through the joint distribution $p_\theta(\boldsymbol{x}, \boldsymbol{z}) = p_\theta(\boldsymbol{x}|\boldsymbol{z})p(\boldsymbol{z})$ where $p(\boldsymbol{z})$ is generally chosen as $\mathcal{N}(\mathbf{0}, \mathbf{I})$. Training the model is done through VI where $q_\varphi(\boldsymbol{z})$ is obtained by explicit amortization over the data point, that is, $q_\varphi(\boldsymbol{z}|\boldsymbol{x}) = \mathcal{N}\left(\boldsymbol{\mu}_\varphi(\boldsymbol{x}), \boldsymbol{\Sigma}_\varphi(\boldsymbol{x})\right)$. Training this system on a dataset $\mathcal{D}$ is done by similarly optimizing the Evidence Lower-Bound, which boils down to the following optimization problem

$$\arg\max_{\theta,\varphi} \mathbb{E}_{\boldsymbol{x}\sim\mathcal{D}}\mathbb{E}_{\boldsymbol{z}\sim q(\cdot|\boldsymbol{x})} \left[\log \frac{p_\theta(\boldsymbol{x}, \boldsymbol{z})}{q_\varphi(\boldsymbol{z}|\boldsymbol{x})}\right] \tag{10}$$

This objective can easily be optimized using gradient-based learning and the reparameterization trick. While typically, a diagonal Gaussian distribution is considered for $q_\varphi$, more complex distributions utilizing normalizing flows can also be used.

### A.2    HYPERNETWORKS

Hypernetworks are neural networks that generate weights for another neural network, used in tasks such as uncertainty quantification, zero-shot learning, etc. We refer for a comprehensive overview to Chauhan et al. (2023). Based on experiments on predicting the weights of a compact MLP (section 4), our work shows similarities with studies in this area but also has significant differences. Regarding uncertainty quantification, hypernetworks are instrumental in creating an ensemble of models by generating multiple weight vectors for the primary network. Each model within this ensemble possesses distinct parameter configurations, enabling robust estimation of uncertainty in model predictions. This feature is precious in safety-critical domains like healthcare, where confidence in predictions is essential. Multiple weight sets can be generated through techniques like dropout within hypernetworks or sampling from a noise distribution. The latter (Krueger et al., 2017) is based on a Bayesian framework where weights can be sampled using invertible network architecture, such as normalizing flows. However, while we amortize posterior inference, the weights sampled from the hypernetwork are not conditioned on information from the currently observed input data during inference time but indirectly solely on the dataset available during training, and retraining would need to be done given a new dataset. Departing from the Bayesian framework, Sun et al. (2017) have shown data-specific discriminative weight prediction, which aligns well with their specific objective of defending a convolutional neural network against adversarial attacks. Combining the ability to sample a new set of weights dataset-specifically but also handling dataset exchangeability, even in the more realistic case of missing information, our work has a distinctly different focus but also can be seen as an extension to hypernetwork research.

### A.3    IN-CONTEXT LEARNING

Amortized inference has close links to in-context learning (ICL), which has been gaining popularity, especially in natural language modeling. Various works show how in-context learning can be seen as performing implicit optimization based on the context examples, with some constructions showing exact equivalence with gradient descent in linear regression (Von Oswald et al., 2023; von Oswald et al., 2023). Other works have shown how such systems can be seen as implicitly modeling the Bayesian posterior predictive distribution (Müller et al., 2021). In a similar vein, there have been additional works aimed at directly modeling the posterior predictive distribution by providing the training data as "context" to a Transformer model and training it based on the maximum log-likelihood

principle (Hollmann et al., 2022). While such approaches have been seeing tremendous success, they cannot be directly applied to cases where we care about and want to analyze the solution space as the solution space is only modeled implicitly, and thus, recovering it is not possible. For example, if our goal is to learn a linear regression model, an ICL model could end up learning a nonlinear model and would provide no information about the actual parameters used for prediction. As opposed to this, we obtain parameters explicitly. We thus can answer questions like the relevance of a particular feature (which corresponds to its weight in the output, and we know the weight vector explicitly). Even further, many systems grounded in physics and economics only admit a constrained solution space; for example, the movement of a human arm lies on a particular manifold, or the configuration of molecules and proteins cannot be arbitrary. Thus, performing predictions through an implicit solution space, which may violate several constraints, is not ideal. Furthermore, explicitly modeling the solution space and encoding the constraints present can be done through the prior and the parametric distribution used for modeling.

## A.4 META LEARNING

Meta-learning (Hospedales et al., 2022) aims to equip models with the ability to quickly learn from different tasks or data sets to generalize to new tasks in resource-constrained domains. This attribute is precious in practical scenarios where obtaining large amounts of task-specific data is impractical or costly. A simple way of obtaining this is through nonparametric or similarity-based models like k-Nearest Neighbours, where no training is involved. Thus, new tasks can be solved quickly based on a few examples by computing a similarity metric with these examples (Koch et al., 2015; Vinyals et al., 2016; Sung et al., 2018). Another way of achieving this is through optimization-based setups, which use a nested optimization procedure. An inner step learns individual tasks from a shared initialization, whereas the outer loop computes the gradient of the whole inner process and moves the initialization in a way that allows for better generalization. Here, by relying on only a few iterations in the inner loop, the outer loop has the incentive to move the initialization to a point from which solutions to multiple tasks are reachable (Finn et al., 2017). Given the similarities between meta-learning and hierarchical Bayesian inference (Grant et al., 2018), our approach can be considered as a kind of meta-learning framework; however, the line between meta-learning and Bayesian posterior inference is quite blurry as any amortized approach for the latter can be seen as a case of the former.

## A.5 NEURAL PROCESSES

A notable approach in meta-learning related to our research is neural processes (NP), which excel in learning scenarios with few examples. NPs (Garnelo et al., 2018a;b; Kim et al., 2019; Pakman et al., 2020; Gordon et al., 2019) can be seen as a more flexible and powerful extension of Gaussian processes that leverage a neural network-based encoder-decoder architecture for learning to model a distribution over functions that approximate a stochastic process. However, while we are interested in approximating the posterior distribution over the parameters, NPs are used to approximate the posterior predictive distribution to make predictions based on observed data. Similar to our setup, NPs rely on amortized VI for obtaining the predictive posterior. Still, instead of working with a known probabilistic model, they train the probabilistic model primarily for prediction-based tasks through approaches analogous to variational expectation maximization. Thus, they cannot provide an explicit posterior over the parameters, but they are suitable for tasks where only predictive posteriors are essential, such as those in supervised learning. NPs, in their most basic form, accomplish this by training for the objective:

$$\arg\max_{\boldsymbol{\theta},\varphi} \mathbb{E}_{\mathcal{D}\sim\chi}\mathbb{E}_{\boldsymbol{z}\sim q_\varphi(\cdot|\mathcal{D})}\left[\log\frac{p_{\boldsymbol{\theta}}(\mathcal{D},\boldsymbol{z})}{q_\varphi(\boldsymbol{z}|\mathcal{D})}\right] \tag{11}$$

where $\boldsymbol{z} \in \mathbb{R}^p$ is an arbitrary latent variable often uninterpretable, and the parameters of the probabilistic model $\boldsymbol{\theta}$ do not get a Bayesian treatment. In particular, NPs are more suited to modeling datasets of the form $\mathcal{D} = \{\boldsymbol{x}_i, \boldsymbol{y}_i\}_{i=1}^n$, where all probabilities in Equation 11 are conditioned on the input $\boldsymbol{x}$'s, and only the predictive over $\boldsymbol{y}$'s is modeled, and $p_{\boldsymbol{\theta}}$ is modeled as a Neural Network.

These approaches can be seen as quite related to ICL, where the exchangeable architecture backbone is switched from DeepSets to Transformers. Similar to ICL, they do not provide control over the solution space as they aim to model either the posterior predictive or an arbitrary latent space. While this leads to good predictive performance on various tasks, they cannot be freely applied to problems

that pose certain constraints on the underlying probabilistic model. In such cases, estimating the actual parameters is important to enforce constraints in the parameter space as well as for interpretability, which we already discussed in the ICL section.

## A.6 SIMULATION-BASED INFERENCE

In the case of simulation-based inference (Cranmer et al., 2020), when the likelihood $p(\boldsymbol{x}|\boldsymbol{\theta})$ is intractable, BayesFlow (Radev et al., 2020) and similar methods (Lorch et al., 2022) provide a solution framework to amortize Bayesian inference of parameters in complex models. Starting from the forward KL divergence between the true and approximate posteriors, the resulting objective is to optimize for parameters of the approximate posterior distribution that maximize the posterior probability of data-generating parameters $\boldsymbol{\theta}$ given observed data $\mathcal{D}$ for all $\boldsymbol{\theta}$ and $\mathcal{D}$. Density estimation of the approximate posterior can then be done using the change-of-variables formula and a conditional invertible neural network that parameterizes the approximate posterior distribution.

$$\underset{\varphi}{\arg\min}\, \mathbb{KL}[p(\boldsymbol{\theta}|\mathcal{D})||q_\varphi(\boldsymbol{\theta}|\mathcal{D})] = \underset{\varphi=\{\nu,\psi\}}{\arg\min}\, \mathbb{E}_{(\boldsymbol{\theta},\mathcal{D})\sim p(\boldsymbol{\theta},\mathcal{D})}\left[-\log p_{\boldsymbol{z}}(f_\nu(\boldsymbol{\theta}; h_\psi(\mathcal{D}))) - \log|\det J_{f_\nu}|\right]$$
(12)

Since their goal is to learn a global estimator for the probabilistic mapping from $\mathcal{D}$ to data generating $\boldsymbol{\theta}$, the information about the observed dataset is encoded in the output of a summary network $h_\psi$. It is used as conditional input to the normalizing flow $f_\nu$. Although the likelihood function does not need to be known, the method requires access to paired observations $(\boldsymbol{x}, \boldsymbol{\theta})$ for training, which is sometimes unavailable. This approach is equivalent to the *Forward KL* setup in our experiments when trained with DeepSets and Normalizing Flows. Current research has also leveraged score-based generative models for SBI which can condition on a dataset by learning a score model conditional only on single observations (Geffner et al., 2023).

## A.7 AMORTIZATION IN GAUSSIAN PROCESSES

Gaussian Processes (GPs) define a class of probabilistic models that do enjoy tractable likelihood. However, inference in such systems is slow and sensitive to the choice of kernel function that defines the covariance matrix. Similar to meta learning and neural processes, current research also focuses on estimating the kernel function in GPs by leveraging permutation invariant architectures like transformers (Liu et al., 2020; Simpson et al., 2021; Bitzer et al., 2023). Additionally, often these approaches amortize based on point estimates and are leveraged when considering GPs for regression problems, and it is not straightforward to extend them to classification or unsupervised learning. In contrast, our approach is more general and can work for all problems that define a differentiable likelihood function. Additionally, our approach also approximates the Bayesian posterior distribution over the parameters of interest, as opposed to point estimates.

## A.8 MODE COLLAPSE IN VARIATIONAL INFERENCE

Reverse KL based methods have been widely known to suffer from mode collapse due to the nature of the optimization objective (Bishop & Nasrabadi, 2006), which implies that even if the approximate distribution possesses the ability to represent multiple modes, optimization is often sub-optimal and the distribution ends up covering only a small handful of them. Improving normalizing flow based methods with repulsive terms or through the lens of natural gradient optimization procedure for a mixture approximate distribution (Arenz et al., 2022; Lin et al., 2020) is an important topic of research, and we believe it would be quite an important future work to experimentally validate if they help in learning multi-modality in amortized posterior inference problems that are studied in this work.

## B ARCHITECTURES RESPECTING EXCHANGEABILITY

In this section, we highlight how DeepSets and Transformer models satisfy the dataset exchangeability criteria, which is essential in modeling the posterior distribution over the parameters of any probabilistic model relying on *iid* data.

### B.1   DEEPSETS

DeepSets (Zaheer et al., 2017) operate on arbitrary sets $\mathcal{X} = \{x_1, ...x_N\} \subset \mathbb{R}^d$ of fixed dimensionality $d$ by first mapping each individual element $\boldsymbol{x}_i \in \mathcal{X}$ to some high-dimensional space using a nonlinear transform, which is parameterized as a multi-layered neural network with parameters $\varphi_1$

$$\boldsymbol{z}_i = f_{\varphi_1}(\boldsymbol{x}_i) \tag{13}$$

After having obtained this high-dimensional embedding of each element of the set, it applies an aggregation function $a(\cdot)$, which is a permutation invariant function that maps a set of elements $\mathcal{Z} = \{\boldsymbol{z}_1, ..., \boldsymbol{z}_N\} \in \mathbb{R}^z$ to an element $\boldsymbol{h} \in \mathbb{R}^z$,

$$\boldsymbol{h} = a(\mathcal{Z}) \tag{14}$$

Thus, the outcome does not change under permutations of $\mathcal{Z}$. Finally, another nonlinear transform, parameterized by a multi-layered neural network with parameters $\varphi_2$, is applied to the outcome $\boldsymbol{h}$ to provide the final output.

$$\boldsymbol{o} = g_{\varphi_2}(\boldsymbol{h}) \tag{15}$$

For our experiments, we then use the vector $\boldsymbol{o}$ to predict the parameters of a parametric family of distributions (e.g., Gaussian or Flows) using an additional nonlinear neural network. As an example, for the Gaussian case, we consider the distribution $\mathcal{N}(\cdot|\boldsymbol{\mu}, \boldsymbol{\Sigma})$, where

$$\boldsymbol{\mu} := \boldsymbol{\mu}_{\varphi_3}(\boldsymbol{o}) \quad \text{and} \quad \boldsymbol{\Sigma} := \boldsymbol{\Sigma}_{\varphi_4}(\boldsymbol{o}) \tag{16}$$

which makes $\boldsymbol{\mu}$ implicitly a function of the original input set $\mathcal{X}$. To understand why the posterior distribution modeled in this fashion does not change when the inputs are permuted, let us assume that $\Pi$ is a permutation over the elements of $\mathcal{X}$. If we look at one of the parameters of the posterior distribution, e.g., $\boldsymbol{\mu}$, we can see that

$$\boldsymbol{\mu}(\Pi\mathcal{X}) = \boldsymbol{\mu}_{\varphi_3}\left(g_{\varphi_2}\left(a\left(\{f_{\varphi_1}(\boldsymbol{x}_{\Pi(i)})\}_{i=1}^N\right)\right)\right) \tag{17}$$

$$= \boldsymbol{\mu}_{\varphi_3}\left(g_{\varphi_2}\left(a\left(\{f_{\varphi_1}(\boldsymbol{x}_i)\}_{i=1}^N\right)\right)\right) \tag{18}$$

$$= \boldsymbol{\mu}(\mathcal{X}) \tag{19}$$

which simply follows from the fact that $a(\cdot)$ is a permutation invariant operation, e.g., sum or mean. We can also provide similar reasoning for the other parameters (e.g., $\boldsymbol{\Sigma}$). This shows that DeepSets can be used to model the posterior distribution over parameters of interest as it respects the exchangeability criteria (*iid* observations) assumptions in the data through its permutation invariant structure.

### B.2   TRANSFORMERS

Similarly, we can look at Transformers (Vaswani et al., 2017) as candidates for respecting the exchangeability conditions in the data. In particular, we consider transformer systems without positional encodings and consider an additional [CLS] token, denoted by $\boldsymbol{c} \in \mathbb{R}^d$, to drive the prediction. If we look at the application of a layer of transformer model, it can be broken down into two components.

**Multi-Head Attention**. Given a query vector obtained from $\boldsymbol{c}$ and keys and values coming from our input set $\mathcal{X} \subset \mathbb{R}^d$, we can model the update of the context $\boldsymbol{c}$ as

$$\hat{\boldsymbol{c}}(\mathcal{X}) = \text{Softmax}\left(\boldsymbol{c}^T \boldsymbol{W}_Q \boldsymbol{W}_K^T \boldsymbol{X}^T\right) \boldsymbol{X} \boldsymbol{W}_V \tag{20}$$

where $\boldsymbol{W}_Q \in \mathbb{R}^{d \times k}, \boldsymbol{W}_K \in \mathbb{R}^{d \times k}, \boldsymbol{W}_V \in \mathbb{R}^{d \times k}$ and $\boldsymbol{X} \in \mathbb{R}^{N \times d}$ denotes a certain ordering of the elements in $\mathcal{X}$. Further, $\hat{\boldsymbol{c}}$ is the updated vector after attention, and Softmax is over the rows of $\boldsymbol{X}$. Here, we see that if we were to apply a permutation to the elements in $\boldsymbol{X}$, the outcome would remain the same. In particular

$$\hat{\boldsymbol{c}}(\Pi\boldsymbol{X}) = \text{Softmax}\left(\boldsymbol{c}^T \boldsymbol{W}_Q \boldsymbol{W}_K^T \boldsymbol{X}^T \Pi^T\right) \Pi\boldsymbol{X}\boldsymbol{W}_V \tag{21}$$

$$= \text{Softmax}\left(\boldsymbol{c}^T \boldsymbol{W}_Q \boldsymbol{W}_K^T \boldsymbol{X}^T\right) \Pi^T \Pi\boldsymbol{X}\boldsymbol{W}_V \tag{22}$$

$$= \text{Softmax}\left(\boldsymbol{c}^T \boldsymbol{W}_Q \boldsymbol{W}_K^T \boldsymbol{X}^T\right) \boldsymbol{X}\boldsymbol{W}_V \tag{23}$$

$$= \hat{\boldsymbol{c}}(\boldsymbol{X}) \tag{24}$$

which follows because Softmax is an equivariant function, i.e., applying Softmax on a permutation of columns is equivalent to applying Softmax first and then permuting the columns correspondingly. Thus, we see that the update to the [CLS] token $c$ is permutation invariant. This output is then used independently as input to a multi-layered neural network with residual connections, and the entire process is repeated multiple times without weight sharing to simulate multiple layers. Since all the individual parts are permutation invariant w.r.t permutations on $\mathcal{X}$, the entire setup ends up being permutation invariant. Obtaining the parameters of a parametric family of distribution for posterior estimation then follows the same recipe as DeepSets, with $o$ replaced by $c$.

## C    PROBABILISTIC MODELS

This section details the various candidate probabilistic models used in our experiments for amortized computation of Bayesian posteriors over the parameters. Here, we explain the parameters associated with the probabilistic model over which we want to estimate the posterior and the likelihood and prior that we use for experimentation.

**Mean of Gaussian (GM):** As a proof of concept, we consider the simple setup of estimating the posterior distribution over the mean of a Gaussian distribution $p(\boldsymbol{\mu}|\mathcal{D})$ given some observed data. In this case, prior and likelihood defining the probabilistic model $p(\boldsymbol{x}, \boldsymbol{\theta})$ (with $\boldsymbol{\theta}$ being the mean $\boldsymbol{\mu}$) are given by:

$$p(\boldsymbol{\mu}) = \mathcal{N}\left(\boldsymbol{\mu}|\mathbf{0}, \mathbf{I}\right) \tag{25}$$
$$p(\boldsymbol{x}|\boldsymbol{\mu}) = \mathcal{N}\left(\boldsymbol{x}|\boldsymbol{\mu}, \boldsymbol{\Sigma}\right) \tag{26}$$

and $\boldsymbol{\Sigma}$ is known beforehand and defined as a unit variance matrix.

**Linear Regression (LR):** We then look at the problem of estimating the posterior over the weight vector for Bayesian linear regression given a dataset $p(\boldsymbol{w}, b|\mathcal{D})$, where the underlying model $p(\mathcal{D}, \boldsymbol{\theta})$ is given by:

$$p(\boldsymbol{w}) = \mathcal{N}(\boldsymbol{w}|\mathbf{0}, \mathbf{I}) \tag{27}$$
$$p(b) = \mathcal{N}(b|0, 1) \tag{28}$$
$$p(y|\boldsymbol{x}, \boldsymbol{w}, b) = \mathcal{N}\left(y|\boldsymbol{w}^T \boldsymbol{x} + b, \sigma^2\right), \tag{29}$$

and with $\sigma^2 = 0.25$ known beforehand. Inputs $\boldsymbol{x}$ are generated from $p(\boldsymbol{x}) = \mathcal{U}(-\mathbf{1}, \mathbf{1})$.

**Linear Classification (LC):** We now consider a setting where the true posterior cannot be obtained analytically as the likelihood and prior are not conjugate. In this case, we consider the underlying probabilistic model by:

$$p(\boldsymbol{W}) = \mathcal{N}\left(\boldsymbol{W}|\mathbf{0}, \mathbf{I}\right) \tag{30}$$
$$p(y|\boldsymbol{x}, \boldsymbol{W}) = \text{Categorical}\left(y\left|\frac{1}{\tau}\,\boldsymbol{W}\boldsymbol{x}\right.\right), \tag{31}$$

where $\tau$ is the known temperature term which is kept as $0.1$ to ensure peaky distributions, and $\boldsymbol{x}$ is being generated from $p(\boldsymbol{x}) = \mathcal{U}(-\mathbf{1}, \mathbf{1})$.

**Nonlinear Regression (NLR):** Next, we tackle the more complex problem where the posterior distribution is multi-modal and obtaining multiple modes or even a single good one is challenging. For this, we consider the model as a Bayesian Neural Network (BNN) for regression with fixed hyper-parameters like the number of layers, dimensionality of the hidden layer, etc. Let the BNN denote the function $f_{\boldsymbol{\theta}}$ where $\boldsymbol{\theta}$ are the network parameters such that the estimation problem is to approximate $p(\boldsymbol{\theta}|\mathcal{D})$. Then, for regression, we specify the probabilistic model using:

$$p(\boldsymbol{\theta}) = \mathcal{N}\left(\boldsymbol{\theta}|\mathbf{0}, \mathbf{I}\right) \tag{32}$$
$$p(y|\boldsymbol{x}, \boldsymbol{\theta}) = \mathcal{N}\left(y|f_{\boldsymbol{\theta}}(\boldsymbol{x}), \sigma^2\right), \tag{33}$$

where $\sigma^2 = 0.25$ is a known quantity and $\boldsymbol{x}$ being generated from $p(\boldsymbol{x}) = \mathcal{U}(-\mathbf{1}, \mathbf{1})$.

**Nonlinear Classification (NLC):** Like in Nonlinear Regression, we consider BNNs with fixed hyper-parameters for classification problems with the same estimation task of approximating $p(\boldsymbol{\theta}|\mathcal{D})$.

In this formulation, we consider the probabilistic model as:

$$p(\boldsymbol{\theta}) = \mathcal{N}\left(\boldsymbol{\theta}|\mathbf{0},\mathbf{I}\right) \tag{34}$$

$$p(y|\boldsymbol{x},\boldsymbol{\theta}) = \text{Categorical}\left(y\left|\frac{1}{\tau}\ f_{\boldsymbol{\theta}}(\boldsymbol{x})\right.\right) \tag{35}$$

where $\tau$ is the known temperature term which is kept as $0.1$ to ensure peaky distributions, and $\boldsymbol{x}$ is being generated from $p(\boldsymbol{x}) = \mathcal{U}(-\mathbf{1},\mathbf{1})$.

**Gaussian Mixture Model (GMM):** While we have mostly looked at predictive problems, where the task is to model some predictive variable $y$ conditioned on some input $\boldsymbol{x}$, we now look at a well-known probabilistic model for unsupervised learning, Gaussian Mixture Model (GMM), primarily used to cluster data. Consider a $K$-cluster GMM with:

$$p(\boldsymbol{\mu}_k) = \mathcal{N}\left(\boldsymbol{\mu}_k|\mathbf{0},\mathbf{I}\right) \tag{36}$$

$$p(\boldsymbol{x}|\boldsymbol{\mu}_{1:K}) = \sum_{k=1}^{K}\pi_k\mathcal{N}\left(\boldsymbol{x}|\boldsymbol{\mu}_k,\boldsymbol{\Sigma}_k\right)\ . \tag{37}$$

We assume $\boldsymbol{\Sigma}_k$ and $\pi_k$ to be known and set $\boldsymbol{\Sigma}_k$ to be an identity matrix and the mixing coefficients to be equal, $\pi_k = 1/K$, for all clusters $k$ in our experiments.

# D    METRICS

In this section, we provide details about the metrics considered for the different tasks. We generally look at two main metrics for benchmarking performance: $L_2$ loss and Accuracy. For estimating the mean of a Gaussian distribution, the $L_2$ loss is defined as

$$GM_{L_2} = \mathbb{E}_{\mathcal{D}\sim\chi}\mathbb{E}_{\boldsymbol{\mu}\sim q_\varphi(\cdot|\mathcal{D})}\left[\sum_{i=1}^{N_{\mathcal{D}}}(\boldsymbol{x}_i - \boldsymbol{\mu})^2\right] \tag{38}$$

where $\mathcal{D} = \{\boldsymbol{x}_i\}_{i=1}^{N_{\mathcal{D}}}$. Intuitively, this captures the quality of the estimation of the mean parameter by measuring how far the observations are from it. Lower value implies better estimation of the mean parameter. Similarly, for estimating the means of a Gaussian Mixture Model, we rely on a similar metric but we also find the cluster closest to the observation, which can be defined as

$$GMM_{L_2} = \mathbb{E}_{\mathcal{D}\sim\chi}\mathbb{E}_{\boldsymbol{\mu}_k\sim q_\varphi(\cdot|\mathcal{D})}\left[\sum_{i=1}^{N_{\mathcal{D}}}(\boldsymbol{x}_i - \boldsymbol{\mu}_{\text{Match}(\boldsymbol{x}_i,\{\boldsymbol{\mu}_1,...\boldsymbol{\mu}_K\})})^2\right] \tag{39}$$

$$\text{Match}(\boldsymbol{x},\{\boldsymbol{\mu}_1,...,\boldsymbol{\mu}_K\}) = \arg\min_k(\boldsymbol{x} - \boldsymbol{\mu}_k)^2 \tag{40}$$

which intuitively captures the distance of observations from the cluster closest to them. Next, we define the metric for evaluating (non-)linear regression models as

$$(N-)LR_{L_2} = \mathbb{E}_{\mathcal{D}\sim\chi}\mathbb{E}_{\boldsymbol{\theta}\sim q_\varphi(\cdot|\mathcal{D})}\left[\sum_{i=1}^{N_{\mathcal{D}}}(y_i - \text{Mode}\left[p(y_i|\boldsymbol{x}_i,\boldsymbol{\theta})\right])^2\right] \tag{41}$$

Finally, for the (non-)linear classification setups, we define the accuracy metric as

$$(N-)LC_{Accuracy} = \mathbb{E}_{\mathcal{D}\sim\chi}\mathbb{E}_{\boldsymbol{\theta}\sim q_\varphi(\cdot|\mathcal{D})}\left[\frac{100}{N_{\mathcal{D}}}\times\sum_{i=1}^{N_{\mathcal{D}}}\delta(y_i,\text{Mode}\left[p(y_i|\boldsymbol{x}_i,\boldsymbol{\theta})\right])\right] \tag{42}$$

where $\delta(a,b) = 1$ if and only if $a = b$. Thus this metric captures the accuracy of the posterior predictive distribution. Another metric that we use to test the quality of the posterior is the symmetric KL divergence, defined as

$$\text{Symmetric }\mathbb{KL}(p(\boldsymbol{\theta}||\mathcal{D}),q_\varphi(\boldsymbol{\theta}|\mathcal{D})) = \frac{1}{2}\mathbb{KL}(p(\boldsymbol{\theta}||\mathcal{D})||q_\varphi(\boldsymbol{\theta}|\mathcal{D})) + \frac{1}{2}\mathbb{KL}(q_\varphi(\boldsymbol{\theta}|\mathcal{D})||p(\boldsymbol{\theta}||\mathcal{D})) \tag{43}$$

Additionally, another metric in the predictive space that we use is the expected negative conditional log likelihood (CNLL), which is defined as

$$CNLL = -\mathbb{E}_{q_\varphi(\cdot|\mathcal{D})}\left[\log p(\mathcal{D}|\boldsymbol{\theta})\right] \tag{44}$$

# E  ARCHITECTURE DETAILS

In this section, we outline the two candidate architectures that we consider for the backbone of our amortized variational inference model. We discuss the specifics of the architectures and the hyperparameters used for our experiments.

## E.1  TRANSFORMER

We use a transformer model (Vaswani et al., 2017) as a permutation invariant architecture by removing positional encodings from the setup and using multiple layers of the encoder model. We append the set of observations with a [CLS] token before passing it to the model and use its output embedding to predict the parameters of the variational distribution. Since no positional encodings or causal masking is used in the whole setup, the final embedding of the [CLS] token becomes invariant to permutations in the set of observations, thereby leading to permutation invariance in the parameters of $q_\varphi$.

We use 4 encoder layers with a 256 dimensional attention block and 1024 feed-forward dimensions, with 4 heads in each attention block for our Transformer models to make the number of parameters comparative to the one of the DeepSets model.

## E.2  DEEPSETS

Another framework that can process set-based input is Deep Sets (Zaheer et al., 2017). In our experiments, we used an embedding network that encodes the input into representation space, a mean aggregation operation, which ensures that the representation learned is invariant concerning the set ordering, and a regression network. The latter's output is either used to directly parameterize a diagonal Gaussian or as conditional input to a normalizing flow, representing a summary statistics of the set input.

For DeepSets, we use 4 layers each in the embedding network and the regression network, with a mean aggregation function, ReLU activation functions, and 627 hidden dimensions to make the number of parameters comparable to those in the Transformer model.

## E.3  NORMALIZING FLOWS

Assuming a Gaussian posterior distribution as the approximate often leads to poor results as the true posterior distribution can be far from the Gaussian shape. To allow for more flexible posterior distributions, we use normalizing flows (Kingma & Dhariwal, 2018; Kobyzev et al., 2020; Papamakarios et al., 2021; Rezende & Mohamed, 2015) for approximating $q_\varphi(\boldsymbol{\theta}|\mathcal{D})$ conditioned on the output of the summary network $h_\psi$. Specifically, let $g_\nu : \boldsymbol{z} \mapsto \boldsymbol{\theta}$ be a diffeomorphism parameterized by a conditional invertible neural network (cINN) with network parameters $\nu$ such that $\boldsymbol{\theta} = g_\nu(\boldsymbol{z}; h_\psi(\mathcal{D}))$. With the change-of-variables formula it follows that $p(\boldsymbol{\theta}) = p(\boldsymbol{z}) \left|\det \frac{\partial}{\partial \boldsymbol{z}} g_\nu(\boldsymbol{z}; h_\psi(\mathcal{D}))\right|^{-1} = p(\boldsymbol{z})|\det J_\nu(\boldsymbol{z}; h_\psi(\mathcal{D}))|^{-1}$, where $J_\nu$ is the Jacobian matrix of $g_\nu$. Further, integration by substitution gives us $d\boldsymbol{\theta} = |\det J_\nu(\boldsymbol{z}; h_\psi(\mathcal{D})|d\boldsymbol{z}$ to rewrite the objective from eq. 7 as:

$$\underset{\varphi}{\arg\min} \, \mathbb{KL}[q_\varphi(\boldsymbol{\theta}|\mathcal{D})||p(\boldsymbol{\theta}|\mathcal{D})] \tag{45}$$

$$= \underset{\varphi}{\arg\min} \, \mathbb{E}_{\mathcal{D}\sim\chi}\mathbb{E}_{\boldsymbol{\theta}\sim q_\varphi(\boldsymbol{\theta}|\mathcal{D})} \left[\log q_\varphi(\boldsymbol{\theta}|\mathcal{D}) - \log p(\boldsymbol{\theta}, \mathcal{D})\right] \tag{46}$$

$$= \underset{\varphi=\{\psi,\nu\}}{\arg\min} \, \mathbb{E}_{\mathcal{D}\sim\chi}\mathbb{E}_{\boldsymbol{z}\sim p(\boldsymbol{z})} \left[\log \frac{q_\nu(\boldsymbol{z}|h_\psi(\mathcal{D}))}{|\det J_\nu(\boldsymbol{z}; h_\psi(\mathcal{D}))|} - \log p(g_\nu(\boldsymbol{z}; h_\psi(\mathcal{D})), \mathcal{D})\right] \tag{47}$$

As shown in BayesFlow (Radev et al., 2020), the normalizing flow $g_\nu$ and the summary network $h_\psi$ can be trained simultaneously. The AllInOneBlock coupling block architecture of the FrEIA Python package (Ardizzone et al., 2018), which is very similar to the RNVP style coupling block (Dinh et al., 2017), is used as the basis for the cINN. AllInOneBlock combines the most common architectural components, such as ActNorm, permutation, and affine coupling operations.

For our experiments, 6 coupling blocks define the normalizing flow network, each with a 1 hidden-layered non-linear feed-forward subnetwork with ReLU non-linearity and 128 hidden dimensions.

## F    EXPERIMENTAL DETAILS

Unless specified, we obtain a stream of datasets for all our experiments by simply sampling from the assumed probabilistic model, where the number of observations $n$ is sampled uniformly in the range $[64, 128]$. For efficient mini-batching over datasets with different cardinalities, we sample datasets with maximum cardinality $(128)$ and implement different cardinalities by masking out different numbers of observations for different datasets whenever required.

To evaluate both our proposed approach and the baselines, we compute an average of the predictive performances across 25 different posterior samples for each of the 100 fixed test datasets for all our experiments. That means for our proposed approach, we sample 25 different parameter vectors from the approximate posterior that we obtain. For MCMC, we rely on 25 MCMC samples, and for optimization, we train 25 different parameter vectors where the randomness comes from initialization. For the optimization baseline, we perform a quick hyperparameter search over the space $\{0.01, 003, 0.001, 0.0003, 0.0001, 0.00003\}$ to pick the best learning rate that works for all of the test datasets and then use it to train for 1000 iterations using the Adam optimizer (Kingma & Ba, 2014). For the MCMC baseline, we use the open-sourced implementation of Langevin-based MCMC sampling[2] where we leave a chunk of the starting samples as burn-in and then start accepting samples after a regular interval (to not make them correlated). The details about the burn-in time and the regular interval for acceptance are provided in the corresponding experiments' sections below.

For our proposed approach of amortized inference, we do not consider explicit hyperparameter optimization and simply use a learning rate of $1e\text{-}4$ with the Adam optimizer. For all experiments, we used linear scaling of the KL term in the training objectives as described in (Higgins et al., 2017), which we refer to as warmup. Furthermore, training details for each experiment can be found below.

### F.1    FIXED-DIM

In this section, we provide the experimental details relevant to reproducing the results of Section 4.1. All the models are trained with streaming data from the underlying probabilistic model, such that every iteration of training sees a new set of datasets. Training is done with a batch size of 128, representing the number of datasets seen during one optimization step. Evaluations are done with 25 samples and we ensure that the test datasets used for each probabilistic model are the same across all the compared methods, i.e., baselines, forward KL, and reverse KL. We train the amortized inference model and the forward KL baselines for the following different probabilistic models:

**Mean of Gaussian (GM):** We train the amortization models over $20,000$ iterations for both the 2-dimensional as well as the 100-dimensional setup. We use a linear warmup with 5000 iterations over which the weight of the KL term in our proposed approach scales linearly from 0 to 1. We use an identity covariance matrix for the data-generating process, but it can be easily extended to the case of correlated or diagonal covariance-based Gaussian distributions.

**Gaussian Mixture Model (GMM):** We train the mixture model setup for $200,000$ iterations with $50,000$ iterations of warmup. We mainly experiment with 2-dimensional and 5-dimensional mixture models, with 2 and 5 mixture components for each setup. While we do use an identity covariance matrix for the data-generating process, again, it can be easily extended to other cases.

**Linear Regression (LR):** The amortization models for this setup are trained for $50,000$ iterations with $12,500$ iterations of warmup. The feature dimensions considered for this task are 1 and 100 dimensions, and the predictive variance $\sigma^2$ is assumed to be known and set as 0.25.

**Nonlinear Regression (NLR):** We train the setup for $100,000$ iterations with $25,000$ iterations consisting of warmup. The feature dimensionalities considered are 1-dimensional and 25-dimensional, and training is done with a known predictive variance similar to the LR setup. For the probabilistic model, we consider both a 1-layered and a 2-layered multi-layer perceptron (MLP) network with 32 hidden units in each, and either a RELU or TANH activation function.

**Linear Classification (LC):** We experiment with 2-dimensional and 100-dimensional setups with training done for $50,000$ iterations, out of which $12,500$ are used for warmup. Further, we train for both binary classification as well as a 5-class classification setup.

---

[2]https://github.com/alisiahkoohi/Langevin-dynamics

**Nonlinear Classification (NLC):** We experiment with 2-dimensional and 25-dimensional setups with training done for $100,000$ iterations, out of which $2,5000$ are used for warmup. Further, we train for both binary classification as well as a 5-class classification setup. For the probabilistic model, we consider both a 1-layered and a 2-layered multi-layer perceptron (MLP) network with 32 hidden units in each, and either a RELU or TANH activation function.

## F.2    VARIABLE-DIM

In this section, we provide the experimental details relevant to reproducing the results of Section 4.2. All the models are trained with streaming data from the underlying probabilistic model, such that every iteration of training sees a new set of datasets. Training is done with a batch size of $128$, representing the number of datasets seen during one optimization step. Further, we ensure that the datasets sampled resemble a uniform distribution over the feature dimensions, ranging from 1-dimensional to the maximal dimensional setup. Evaluations are done with 25 samples and we ensure that the test datasets used for each probabilistic model are the same across all the compared methods, i.e., baselines, forward KL, and reverse KL. We train the amortized inference model and the forward KL baselines for the following different probabilistic models:

**Mean of Gaussian (GM):** We train the amortization models over $50,000$ iterations using a linear warmup with $12,5000$ iterations over which the weight of the KL term in our proposed approach scales linearly from 0 to 1. We use an identity covariance matrix for the data-generating process, but it can be easily extended to the case of correlated or diagonal covariance-based Gaussian distributions. In this setup, we consider a maximum of 100 feature dimensions.

**Gaussian Mixture Model (GMM):** We train the mixture model setup for $500,000$ iterations with $125,000$ iterations of warmup. We set the maximal feature dimensions as 5 and experiment with 2 and 5 mixture components. While we do use an identity covariance matrix for the data-generating process, again, it can be easily extended to other cases.

**Linear Regression (LR):** The amortization models for this setup are trained for $100,000$ iterations with $25,000$ iterations of warmup. The maximal feature dimension considered for this task is 100-dimensional, and the predictive variance $\sigma^2$ is assumed to be known and set as $0.25$.

**Nonlinear Regression (NLR):** We train the setup for $250,000$ iterations with $62,500$ iterations consisting of warmup. The maximal feature dimension considered is 100-dimensional, and training is done with a known predictive variance similar to the LR setup. For the probabilistic model, we consider both a 1-layered and a 2-layered multi-layer perceptron (MLP) network with 32 hidden units in each, and either a RELU or TANH activation function.

**Linear Classification (LC):** We experiment with a maximal 100-dimensional setup with training done for $100,000$ iterations, out of which $25,000$ are used for warmup. Further, we train for both binary classification as well as a 5-class classification setup.

**Nonlinear Classification (NLC):** We experiment with a maximal 100-dimensional setup with training done for $250,000$ iterations, out of which $62,500$ are used for warmup. Further, we train for both binary classification as well as a 5-class classification setup. For the probabilistic model, we consider both a 1-layered and a 2-layered multi-layer perceptron (MLP) network with 32 hidden units in each, and either a RELU or TANH activation function.

## F.3    MODEL MISSPECIFICATION

In this section, we provide the experimental details relevant to reproducing the results of Section 4.3. All models during this experiment are trained with streaming data from the currently used dataset-generating function $\chi$, such that every iteration of training sees a new batch of datasets. Training is done with a batch size of $128$, representing the number of datasets seen during one optimization step. Evaluation for all models is done with 10 samples from each dataset-generator used in the respective experimental subsection and we ensure that the test datasets are the same across all compared methods, i.e., baselines, forward KL, and reverse KL.

**Linear Regression Model:** The linear regression amortization models are trained following the training setting for linear regression fixed dimensionality, that is, $50,000$ training iterations with $12,500$ iterations of warmup. The feature dimension considered for this task is 1-dimension. The

model is trained separately on datasets from three different generators $\chi$: linear regression, nonlinear regression, and Gaussian processes, and evaluated after training on test datasets from all of them. For training with datasets from the linear regression probabilistic model, the predictive variance $\sigma^2$ is assumed to be known and set as $0.25$. The same variance is used for generating datasets from the nonlinear regression dataset generator with 1 layer, 32 hidden units, and TANH activation function. Lastly, datasets from the Gaussian process-based generator are sampled similarly, using the GPytorch library Gardner et al. (2018), where datasets are sampled of varying cardinality, ranging from 64 to 128. We use a zero-mean Gaussian Process (GP) with a unit lengthscale radial-basis function (RBF) kernel serving as the covariance matrix. Further, we use a very small noise of $\sigma^2 = 1\mathrm{e}^{-6}$ in the likelihood term of the GP. Forward KL training in this experiment can only be done when the amortization model and the dataset-generating function are the same: when we train on datasets from the linear regression-based $\chi$. Table 13 provides a detailed overview of the results.

**Nonlinear Regression Models:** The nonlinear regression amortization models are trained following the training setting for nonlinear regression fixed dimensionality, that is, $100,000$ training iterations with $25,000$ iterations of warmup. Here, we consider two single-layer perceptions with 32 hidden units and either a RELU or TANH activation function. The feature dimensionality considered is 1 dimension. We consider the same three dataset-generating functions as in the misspecification experiment for a linear regression model above. However, the activation function used in the nonlinear regression dataset generator matches the activation function of the currently trained amortization model. In this case, forward KL training is possible in the two instances when trained on datasets from the corresponding nonlinear regression probabilistic model. A more detailed overview of the results can be found in Table 14 for the TANH and in Table 15 for the RELU activation function-based probabilistic models respectively.

### F.4 TABULAR EXPERIMENTS

For the tabular experiments, we train the amortized inference models for (non-)linear regression (NLR/LR) as well as (non-)linear classification (NLC/LC) with $x \sim \mathcal{N}(\mathbf{0}, \mathbf{I})$ as opposed to $x \sim \mathcal{U}(-\mathbf{1}, \mathbf{1})$ in the dataset generating process $\chi$, with the rest of the settings the same as MAXIMUM-DIM experiments. For the nonlinear setups, we only consider the RELU case as it has seen predominant success in deep learning. Further, we only consider a 1-hidden layer neural network with 32 hidden dimensions in the probabilistic model.

After having trained the amortized inference models, both for forward and reverse KL setups, we evaluate them on real-world tabular datasets. We first collect a subset of tabular datasets from the OpenML platform as outlined in Appendix G. Then, for each dataset, we perform a 5-fold cross-validation evaluation where the dataset is chunked into 5 bins, of which, at any time, 4 are used for training and one for evaluation. This procedure is repeated five times so that every chunk is used for evaluation once.

For each dataset, we normalize the observations and the targets so that they have zero mean and unit standard deviation. For the classification setups, we only normalize the inputs as the targets are categorical. For both forward KL and reverse KL amortization models, we initialize the probabilistic model from samples from the amortized model and then further finetune it via dataset-specific maximum a posteriori optimization. We repeat this setup over 25 different samples from the inference model. In contrast, for the optimization baseline, we initialize the probabilistic models' parameters from $\mathcal{N}(0, I)$, which is the prior that we consider, and then train 25 such models with maximum a posteriori objective using Adam optimizer.

While we see that the amortization models, particularly the reverse KL model, lead to much better initialization and convergence, it is important to note that the benefits vanish if we initialize using the Xavier-init initialization scheme. However, we believe that this is not a fair comparison as it means that we are considering a different prior now, while the amortized models were trained with $\mathcal{N}(0, I)$ prior. We defer the readers to the section below for additional discussion and experimental results.

## G OPENML DATASETS

For the tabular regression problems, we consider the suite of tasks outlined in *OpenML-CTR23 - A curated tabular regression benchmarking suite* (Fischer et al., 2023), from which we further filter

out datasets that have more than 2000 examples and 100 features. We also remove datasets with missing information and NaNs. Similarly, we consider the *OpenML-CC18 Curated Classification benchmark* (Bischl et al., 2019) suite of tasks for classification and perform a similar filtering algorithm. We remove datasets with missing information and NaNs, as well as datasets with more than 2000 examples and 100 features. In addition, we also exclude datasets that are not made for binary classification. At the end of this filtering mechanism, we end up with 9 regression and 13 classification problems, and our dataset filtration pipeline is heavily inspired by Hollmann et al. (2022). We provide the datasets considered for both regression and classification below:

**Regression**: AIRFOIL_SELF_NOISE, CONCRETE_COMPRESSIVE_STRENGTH, ENERGY_EFFICIENCY, SOLAR_FLARE, STUDENT_PERFORMANCE_POR, QSAR_FISH_TOXICITY, RED_WINE, SOCMOB and CARS.

**Classification**: CREDIT-G, DIABETES, TIC-TAC-TOE, PC4, PC3, KC2, PC1, BANKNOTE-AUTHENTICATION, BLOOD-TRANSFUSION-SERVICE-CENTER, ILPD, QSAR-BIODEG, WDBC and CLIMATE-MODEL-SIMULATION-CRASHES.

## H  ADDITIONAL EXPERIMENTS

In this section, we outline the additional experiments we conducted in obtaining Bayesian posteriors for the different probabilistic models for different hyperparameters and their downstream uses. We provide a comprehensive account of the results in the relevant sections below.

### H.1  FIXED-DIM

While we highlighted the results with the Gaussian mixture model and classification settings with only 2 clusters/classes, we also conducted experiments with an increased number of clusters and classes, making the problem even more challenging. Table 7 shows that both forward and reverse KL methods perform reasonably, with forward KL struggling more in challenging scenarios.

Next, we also consider harder tasks based on the Bayesian Neural Network (BNN) paradigm, where we consider nonlinear regression and classification setups with different activation functions: TANH and RELU for a 1-layered and 2-layered BNN. We provide the results of our experiments in Tables 8 and 9 respectively. The results indicate that forward KL approaches struggle a lot in such scenarios, often achieving performance comparable to random chance. On the contrary, we see that reverse KL-based amortization leads to performances often similar to dataset-specific optimization, thereby showing the superiority of our proposed method.

### H.2  VARIABLE-DIM

Our experiments on variable dimensional datasets can be evaluated for arbitrary feature cardinality, of which we show a few examples in Section 4.2. In this section, we provide results for additional dimensionality setups. In particular, we refer the readers to Table 10, which contains experimental results w.r.t different dimensionalities (e.g. 50D setup), as well as different number of clusters and classes, respectively, for the GMM and LC setup. Throughout, we see that amortization leads to reasonable performance, and in particular, we see forward KL-based amortization starting to struggle in high-dimensional setups.

Again, to make the setup more challenging, we consider the Bayesian Neural Network (BNN) setup where we consider nonlinear regression and classification with different activation functions: TANH and RELU for a 1-layered and 2-layered BNN, but which can now be tested for an arbitrary number of input features. Our experiments are highlighted in Tables 11 and 12, for 1- and 2-layered BNN, respectively. In such complex multi-modal and complicated setups, forward KL often performs comparable to random chance and thus does not lead to any good approximation of the true posterior distribution. On the other hand, our proposed method indeed leads to good predictive performance, often comparable to dataset-specific optimization routines.

## H.3    Model Misspecification

As a representative of the results on model misspecification (Section 4.3), we highlighted training and evaluation of the amortization models with Transformer backbone on a subset of in-distribution and OoD data-generating functions (Table 3) to show superiority in generalization of reverse KL trained system vs. forward KL based ones on OoD data but also to highlight that training a misspecified amortization model on OoD datasets directly with our approach results in even better posterior predictive performance.

In addition to those experiments, we also conducted a broader range of experiments utilizing DeepSets as the backbone, various OoD data-generating functions for training and evaluation of the reverse KL system, and an additional nonlinear regression model with RELU activation function. For a comprehensive description of these experiments and the complete setup, please refer to Section F.3. We considered three probabilistic models, including a linear regression model and two nonlinear regression models utilizing the TANH or RELU activation function. The detailed results for each model can be found in Tables 13, 14, and 15, respectively.

In all experiments, reverse KL outperforms forward KL trained amortization models in in-distribution performance and excels in posterior prediction on OoD datasets. Although the significant difference in posterior prediction performance of forward vs. reverse KL in cases where the underlying model is nonlinear was already mentioned in previous experiments, here, reverse KL-trained models also excel in evaluations of posterior prediction for the linear regression model. Although only by a margin, in the case of approximating the posterior of the simpler linear regression model, a diagonal Gaussian-shaped posterior shows the best posterior prediction results when evaluated on OoD datasets from the nonlinear regression dataset generating function. In almost all other experiments, the posterior prediction performance could be enhanced when we used the normalizing flow based posterior. A definitive conclusion cannot be drawn regarding the superiority of one backbone over the other, i.e. between DeepSets or Transformer. However, amortization models with DeepSets as the backbone tend towards better generalization regarding OoD datasets.

## H.4    Tabular Experiments

As a case of extreme OoD generalization, we test our amortized models trained to handle variable feature dimensions on the suite of regression and classification problems that we filtered out from the OpenML platform, as outlined in Appendix G. We consider both linear and nonlinear probabilistic models to tackle the regression and binary classification setups, which lead to predicting the parameters of a linear regression/classification model and a small nonlinear neural network based on RELU activation function. Further, we also perform the analysis with a diagonal Gaussian assumption and a normalizing flow-based amortization model trained with both a forward and reverse KL objective. We provide the results on the regression problems in (a) linear model with diagonal Gaussian assumption (Figure 8), (b) linear model with normalizing flow (Figure 9), (c) nonlinear model with diagonal Gaussian assumption (Figure 10), and (d) nonlinear model with normalizing flow (Figure 11). The results of the classification problems are shown in (a) linear model with diagonal Gaussian assumption (Figure 12), (b) linear model with normalizing flow (Figure 13), (c) nonlinear model with diagonal Gaussian assumption (Figure 14), and (d) nonlinear model with normalizing flow (Figure 15). Our experiments indicate that initializing with amortized models leads to better performance and training than models trained via maximum a-posteriori approach and initialized with the prior, i.e., $\mathcal{N}(0, I)$.

We do provide an additional baseline of initializing with XAVIER-INIT initialization, which often leads to faster convergence; however, as we consider the prior to be a unit normal, this is an unfair baseline as we assume the weights to be initialized from a different prior. We leave the work of computing Bayesian posteriors with different priors and testing an amortized Bayesian model with XAVIER-INIT prior for the future.

| $q_\varphi$ | | Model | GM | | GMM | LR | | NLR | | LC | | NLC | |
|---|---|---|---|---|---|---|---|---|---|---|---|---|---|
| | | | *CNLL* (↓) | | | | | | | | | | |
| | | | *2D* | *100D* | *5D 2 cl* | *1D* | *100D* | *1D* | *25D* | *2D* | *100D* | *2D* | *25D* |
| Baseline | - | Random | 437.7 | 22581.2 | 3572.8 | 566.6 | 12967.8 | 7759.1 | 53006.3 | 78.5 | 311.7 | 172.9 | 600.4 |
| | - | Optimization | 264.5 | 13295.9 | 193.7 | 69.1 | 1433.9 | 75.5 | 5604.2 | 15.0 | 128.5 | 10.0 | 81.3 |
| | - | MCMC | 267.4 | 13543.3 | 266.7 | 73.3 | 1990.6 | N/A | 7277.8 | 20.1 | 382.6 | 18.5 | 1094.2 |
| Fwd-KL | Gaussian | DeepSets | 265.7 | 13403.9 | 1574.8 | 70.3 | 9749.5 | 6119.1 | 45516.3 | 42.6 | 313.4 | 140.6 | 510.3 |
| | | Transformer | 265.6 | 13387.3 | 1576.6 | 70.2 | 3669.1 | 6281.3 | 43716.3 | 43.1 | 212.8 | 138.2 | 510.5 |
| Rev-KL | | DeepSets | 265.6 | 13372.5 | 239.7 | 70.1 | 4826.8 | 86.6 | 7976.5 | 20.3 | 186.8 | 24.4 | 95.4 |
| | | Transformer | 265.6 | 13357.9 | 250.4 | 70.4 | 2126.5 | 86.6 | 5808.7 | 20.3 | 174.4 | 24.8 | 126.3 |
| Fwd-KL | Flow | DeepSets | 265.7 | 13409.9 | 1113.4 | 70.3 | 9894.3 | 2154.9 | 35493.8 | 37.3 | 311.6 | 115.5 | 423.0 |
| | | Transformer | 265.6 | 13386.6 | 615.6 | 70.4 | 3806.0 | 2331.2 | 34746.5 | 36.8 | 182.6 | 85.7 | 419.0 |
| Rev-KL | | DeepSets | 265.5 | 13368.6 | 256.5 | 70.2 | 5478.9 | 83.2 | 13995.1 | 19.6 | 146.7 | 22.5 | 75.4 |
| | | Transformer | 265.8 | 13364.1 | 223.4 | 70.3 | 2030.2 | 82.7 | 5804.6 | 20.0 | 145.3 | 21.3 | 91.8 |

Table 5: **Fixed-Dimension Posterior Prediction:** Experimental results for posterior inference on fixed dimensional datasets evaluated on estimating the (a) mean of a Gaussian (GM), (b) means of Gaussian mixture model (GMM), (c) parameters for (non-)linear regression (NLR/LR), and (d) parameters for (non-)linear classification (NLC/LC). We consider different backbone architectures and parametric distributions $q_\varphi$, and use dataset-specific Bayesian and point estimates as baselines. CNLL refers to the negative of the expected conditional log likelihood.

| | $q_\varphi$ | Model | $CNLL$ (↓) | | | | | | | | | |
|---|---|---|---|---|---|---|---|---|---|---|---|---|
| | | | GM | GMM | LR | | NLR | | LC | | NLC | |
| | | | 100D | 5D 2 cl | 1D | 100D | 1D | 50D | 2D | 100D | 2D | 50D |
| Baseline | - | Random | 23118.9 | 3462.5 | 581.8 | 13407.0 | 7553.7 | 103462.2 | 77.6 | 321.6 | 178.6 | 865.7 |
| | - | Optimization | 13630.0 | 200.4 | 69.5 | 1350.3 | 79.9 | 18012.8 | 17.7 | 125.2 | 12.3 | 77.6 |
| | - | MCMC | 14019.9 | 402.7 | 91.0 | 2267.1 | N/A | 19443.9 | 29.9 | 298.2 | 41.3 | 2533.6 |
| Fwd-KL | Gaussian | DeepSets | 15641.7 | 1550.4 | 71.8 | 10806.3 | 5471.5 | 88819.0 | 43.6 | 315.5 | 137.2 | 709.7 |
| | | Transformer | 14105.9 | 1580.8 | 73.1 | 4373.6 | 5642.0 | 86694.0 | 47.1 | 216.1 | 134.2 | 707.5 |
| Rev-KL | | DeepSets | 13839.4 | 224.6 | 72.0 | 4707.8 | 129.3 | 28394.0 | 24.3 | 187.7 | 25.1 | 96.4 |
| | | Transformer | 13819.6 | 221.2 | 71.3 | 2233.0 | 124.5 | 15669.6 | 23.2 | 173.5 | 25.7 | 215.1 |
| Fwd-KL | Flow | DeepSets | 15495.9 | 822.8 | 71.5 | 11447.5 | 4607.5 | 72458.0 | 39.8 | 314.2 | 123.6 | 603.0 |
| | | Transformer | 14064.6 | 226.8 | 71.8 | 4649.2 | 3960.3 | 70083.0 | 40.4 | 192.9 | 122.3 | 596.3 |
| Rev-KL | | DeepSets | 14048.4 | 253.3 | 71.5 | 5563.3 | 128.3 | 28703.6 | 22.6 | 145.4 | 28.6 | 77.4 |
| | | Transformer | 13829.4 | 217.8 | 71.0 | 2378.3 | 110.2 | 16439.6 | 22.8 | 143.2 | 24.7 | 126.7 |

Table 6: **Variable-Dimension Posterior Prediction:** Experimental results for posterior inference on variable dimensional datasets evaluated on estimating the (a) mean of a Gaussian (GM), (b) means of Gaussian mixture model (GMM), (c) parameters for (non-)linear regression (NLR/LR), and (d) parameters for (non-)linear classification (NLC/LC). We consider different backbone architectures and parametric distributions $q_\varphi$, and use dataset-specific Bayesian and point estimates as baselines. CNLL refers to the negative of the expected conditional log likelihood.

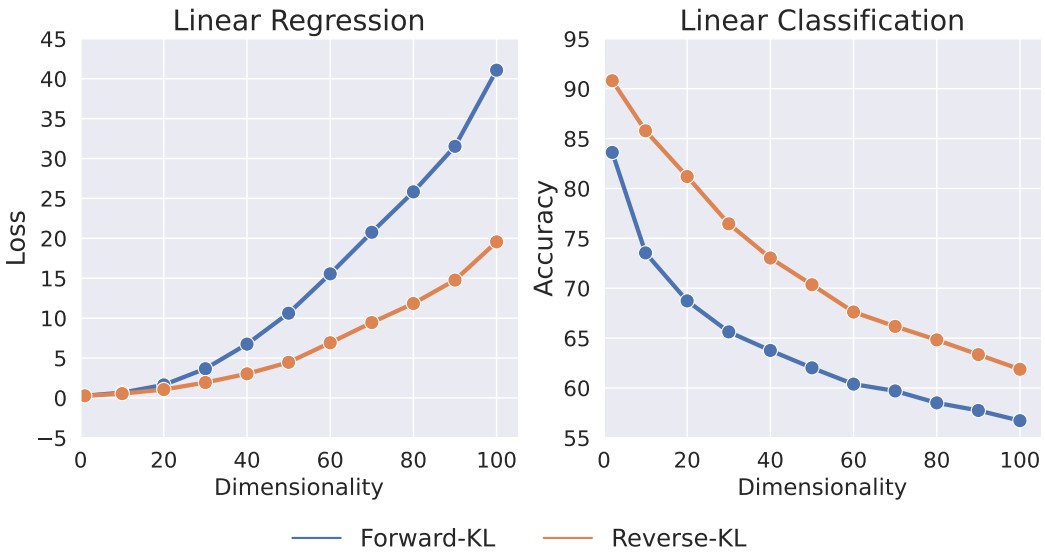

Figure 5: **Trends of Performance over different Dimensions in Variable Dimensionality Setup:** We see that our proposed reverse KL methodology outperforms the forward KL one.

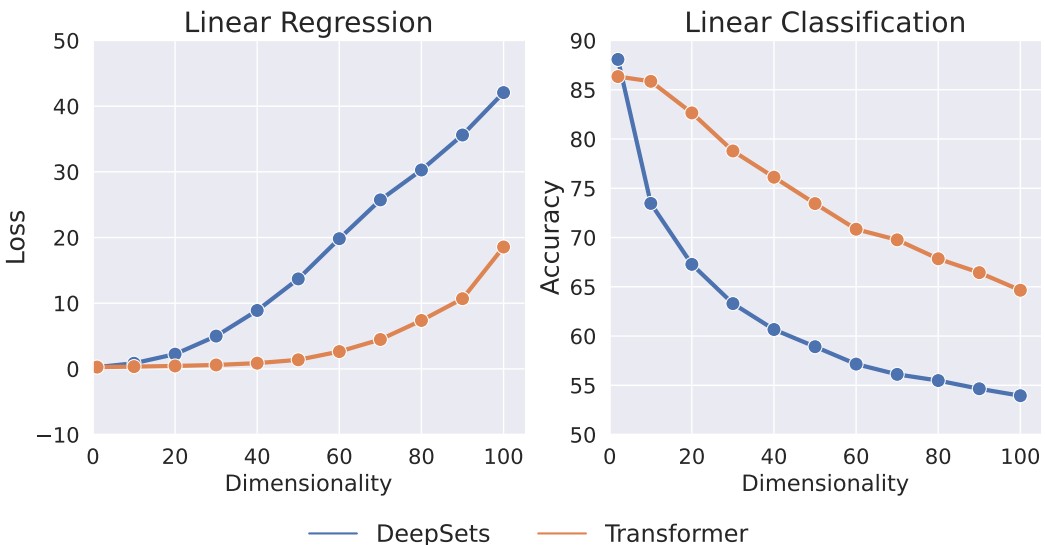

Figure 6: **Trends of Performance over different Dimensions in Variable Dimensionality Setup:** We see that transformer models generalize better to different dimensional inputs than DeepSets.

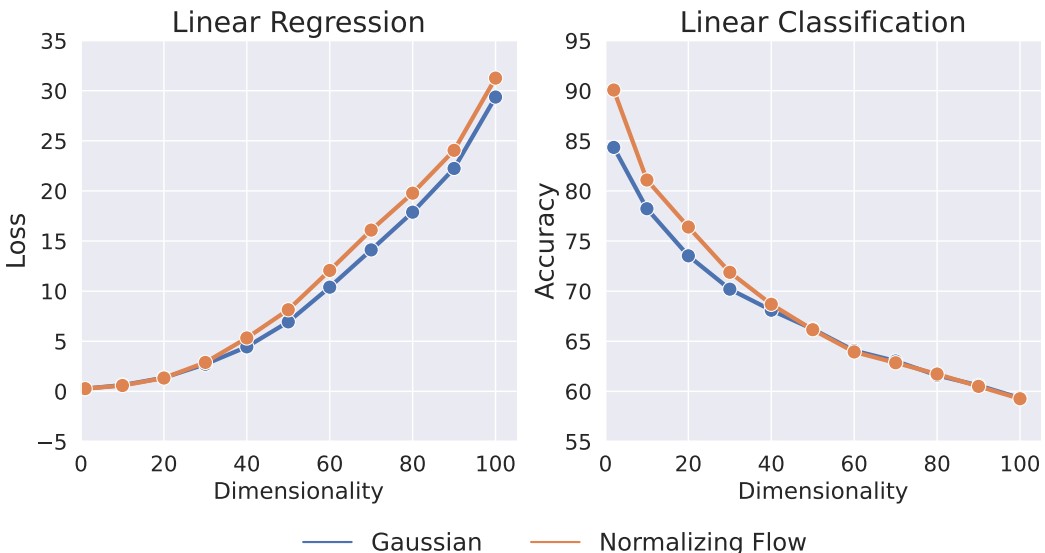

Figure 7: **Trends of Performance over different Dimensions in Variable Dimensionality Setup:** We see that normalizing flows leads to similar performances than Gaussian based variational approximation.

| | $q_\varphi$ | Model | $L_2$ *Loss* ($\downarrow$) GMM | | | *Accuracy* ($\uparrow$) LC | |
|---|---|---|---|---|---|---|---|
| | | | *2D-2cl* | *2D-5cl* | *5D-5cl* | *2D-5cl* | *100D-5cl* |
| Baseline | - | Prior | 1.92 | 0.72 | 5.14 | 20.52 | 19.97 |
| | - | Optimization | 0.17 | 0.12 | 0.43 | 84.75 | 41.55 |
| | - | MCMC | 0.18 | 0.13 | 0.58 | 76.50 | 29.95 |
| Fwd-KL | Gaussian | DeepSets | 0.91 | 0.54 | 2.44 | 66.57 | 19.92 |
| | | Transformer | 0.93 | 0.54 | 2.46 | 68.22 | 26.12 |
| Rev-KL | | DeepSets | 0.18 | 0.13 | 0.47 | 80.91 | 23.94 |
| | | Transformer | 0.20 | 0.13 | 0.46 | 80.96 | 29.95 |
| Fwd-KL | Flow | DeepSets | 0.19 | 0.23 | 0.61 | 81.72 | 20.12 |
| | | Transformer | 0.20 | 0.26 | 0.68 | 82.11 | 26.58 |
| Rev-KL | | DeepSets | 0.18 | 0.13 | 0.51 | 81.48 | 20.39 |
| | | Transformer | 0.18 | 0.13 | 0.52 | 81.46 | 30.63 |

Table 7: **Fixed-Dim Posterior Prediction:** Experimental results for posterior inference on fixed dimensional datasets evaluated on estimating the (a) means of Gaussian mixture model (GMM), and (b) parameters for linear classification (LC) for additional probabilistic model setups (eg. multi-class). We consider different backbone architectures and parametric distributions $q_\varphi$, and use dataset-specific Bayesian and point estimates as baselines. $L_2$ Loss and Accuracy refer to the expected posterior-predictive $L_2$ loss and accuracy respectively. Here, cl refers to the number of clusters for GMM and number of classes for LC.

| Setup | $q_\varphi$ | Model | $L_2$ Loss (↓) NLR | | Accuracy (↑) NLC | | | |
|---|---|---|---|---|---|---|---|---|
| | | | 1D | 25D | 2D-2cl | 2D-5cl | 25D-2cl | 25D-5cl |
| TANH | Baseline | - Prior | 31.47 | 47.37 | 50.31 | 19.88 | 49.72 | 20.08 |
| | | - Optimization | 0.27 | 8.64 | 97.77 | 94.03 | 78.21 | 54.05 |
| | | - MCMC | 0.28 | 12.08 | 96.98 | 90.84 | 66.87 | 37.25 |
| | Fwd-KL (Gaussian) | DeepSets | 30.22 | 47.17 | 49.99 | 19.22 | 49.89 | 19.71 |
| | | Transformer | 30.32 | 47.15 | 49.98 | 19.45 | 49.89 | 19.85 |
| | Rev-KL (Gaussian) | DeepSets | 0.38 | 9.87 | 92.39 | 78.09 | 49.86 | 19.70 |
| | | Transformer | 0.38 | 8.81 | 92.82 | 78.61 | 73.27 | 19.71 |
| | Fwd-KL (Flow) | DeepSets | 31.18 | 45.87 | 49.91 | 19.96 | 49.95 | 19.96 |
| | | Transformer | 13.29 | 46.37 | 49.93 | 19.95 | 49.95 | 20.06 |
| | Rev-KL (Flow) | DeepSets | 0.37 | 21.52 | 93.04 | 82.00 | 49.99 | 19.98 |
| | | Transformer | 0.36 | 8.54 | 93.06 | 81.96 | 50.03 | 20.03 |
| RELU | Baseline | - Prior | 42.65 | 289.83 | 49.89 | 19.93 | 49.80 | 19.62 |
| | | - Optimization | 0.29 | 30.52 | 96.56 | 94.51 | 79.05 | 60.01 |
| | | - MCMC | N/A | 39.57 | 95.81 | 92.18 | 72.05 | 46.62 |
| | Fwd-KL (Gaussian) | DeepSets | 31.24 | 243.62 | 59.22 | 32.08 | 57.71 | 30.48 |
| | | Transformer | 32.06 | 233.75 | 60.09 | 32.73 | 57.57 | 30.81 |
| | Rev-KL (Gaussian) | DeepSets | 0.35 | 43.37 | 90.52 | 82.92 | 60.41 | 34.28 |
| | | Transformer | 0.35 | 31.42 | 90.34 | 84.13 | 74.86 | 45.57 |
| | Fwd-KL (Flow) | DeepSets | 11.46 | 186.95 | 61.57 | 35.17 | 58.52 | 31.91 |
| | | Transformer | 12.61 | 182.98 | 69.53 | 35.68 | 58.43 | 32.08 |
| | Rev-KL (Flow) | DeepSets | 0.33 | 74.97 | 90.87 | 84.46 | 61.05 | 34.74 |
| | | Transformer | 0.33 | 31.30 | 91.51 | 84.72 | 75.11 | 45.23 |

Table 8: **Fixed-Dim Posterior Prediction:** Experimental results for posterior inference on fixed dimensional datasets evaluated on estimating the parameters of nonlinear regression (NLR) and classification (NLC) setups, with 1 layered MLP with different activation functions in the probabilistic model. We also consider a multi-class classification setup. We consider different backbone architectures and parametric distributions $q_\varphi$, and use dataset-specific Bayesian and point estimates as baselines. $L_2$ Loss and Accuracy refer to the expected posterior-predictive $L_2$ loss and accuracy respectively. Here, cl refers to the number classes.

| Setup | $q_\varphi$ | Model | $L_2$ Loss ($\downarrow$) NLR | | Accuracy ($\uparrow$) NLC | | | |
|---|---|---|---|---|---|---|---|---|
| | | | 1D | 25D | 2D-2cl | 2D-5cl | 25D-2cl | 25D-5cl |
| TANH | Baseline | - | Prior | 53.78 | 54.84 | 49.76 | 19.48 | 50.00 | 20.07 |
| | | - | Optimization | 0.48 | 26.95 | 97.25 | 91.81 | 69.63 | 42.20 |
| | | - | MCMC | 0.34 | 29.80 | 95.09 | 84.68 | 52.27 | 24.28 |
| | Fwd-KL (Gaussian) | | DeepSets | 54.58 | 55.63 | 50.03 | 19.75 | 50.11 | 20.09 |
| | | | Transformer | 54.36 | 55.95 | 50.03 | 19.97 | 50.11 | 20.24 |
| | Rev-KL (Gaussian) | | DeepSets | 0.70 | 26.82 | 84.48 | 66.21 | 50.12 | 20.06 |
| | | | Transformer | 0.71 | 16.73 | 84.04 | 66.54 | 50.10 | 20.10 |
| | Fwd-KL (Flow) | | DeepSets | 52.97 | 51.39 | 49.77 | 19.89 | 49.82 | 19.96 |
| | | | Transformer | 52.58 | 51.78 | 49.81 | 20.06 | 49.92 | 20.38 |
| | Rev-KL (Flow) | | DeepSets | 0.66 | 24.19 | 86.46 | 42.63 | 49.42 | 19.90 |
| | | | Transformer | 0.64 | 15.98 | 86.03 | 68.84 | 49.46 | 20.16 |
| RELU | | | Prior | 752.06 | 4846.76 | 49.12 | 19.91 | 50.10 | 19.79 |
| | | | Optimization | 1.39 | 609.99 | 98.08 | 96.91 | 80.72 | 60.15 |
| | | | MCMC | N/A | N/A | 84.43 | 48.73 | 64.77 | 32.29 |
| | Fwd-KL (Gaussian) | | DeepSets | 564.45 | 3995.57 | 57.51 | 31.73 | 58.79 | 30.07 |
| | | | Transformer | 569.48 | 4087.63 | 58.04 | 32.37 | 58.53 | 29.94 |
| | Rev-KL (Gaussian) | | DeepSets | 0.87 | 765.99 | 89.49 | 72.16 | 66.97 | 43.43 |
| | | | Transformer | 0.80 | 611.34 | 91.18 | 78.09 | 67.19 | 44.39 |
| | Fwd-KL (Flow) | | DeepSets | 528.56 | 2584.34 | 57.66 | 32.93 | 66.60 | 30.60 |
| | | | Transformer | 529.59 | 2605.93 | 58.76 | 33.36 | 66.92 | 30.75 |
| | Rev-KL (Flow) | | DeepSets | 0.87 | 732.04 | 89.95 | 72.49 | 77.29 | 45.59 |
| | | | Transformer | 0.68 | 484.93 | 90.71 | 81.36 | 77.01 | 45.14 |

Table 9: **Fixed-Dim Posterior Prediction:** Experimental results for posterior inference on fixed dimensional datasets evaluated on estimating the parameters of nonlinear regression (NLR) and classification (NLC) setups, with 2 layered MLP with different activation functions in the probabilistic model. We also consider a multi-class classification setup. We consider different backbone architectures and parametric distributions $q_\varphi$, and use dataset-specific Bayesian and point estimates as baselines. $L_2$ Loss and Accuracy refer to the expected posterior-predictive $L_2$ loss and accuracy respectively. Here, cl refers to the number classes.

| | $q_\varphi$ | Model | $L_2$ Loss ($\downarrow$) | | | | | Accuracy ($\uparrow$) | | | |
|---|---|---|---|---|---|---|---|---|---|---|---|
| | | | GM | GMM | | | LR | LC | | | |
| | | | 50D | 2D-2cl | 2D-5cl | 5D-5cl | 50D | 2D-5cl | 50D-2cl | 50D-5cl | 100D-5cl |
| Baseline | - | Prior | 153.50 | 3.33 | 0.91 | 1.64 | 35.93 | 19.95 | 49.99 | 20.06 | 20.10 |
| | - | Optimization | 50.51 | 0.21 | 0.13 | 0.33 | 0.63 | 85.15 | 79.93 | 52.32 | 42.21 |
| Fwd-KL | Gaussian | DeepSets | 52.16 | 2.44 | 0.74 | 1.22 | 18.94 | 20.51 | 51.53 | 20.05 | 20.07 |
| | | Transformer | 51.68 | 2.42 | 0.74 | 1.22 | 1.53 | 59.08 | 69.98 | 39.54 | 26.50 |
| Rev-KL | | DeepSets | 51.28 | 0.94 | 0.37 | 0.39 | 7.51 | 79.97 | 68.20 | 32.07 | 25.38 |
| | | Transformer | 51.19 | 0.21 | 0.32 | 0.32 | 1.42 | 80.29 | 73.21 | 42.14 | 30.91 |
| Fwd-KL | Flow | DeepSets | 52.27 | 1.51 | 0.46 | 0.51 | 22.71 | 20.46 | 51.53 | 19.93 | 19.99 |
| | | Transformer | 51.81 | 1.55 | 0.52 | 0.58 | 1.62 | 73.40 | 73.90 | 40.90 | 26.32 |
| Rev-KL | | DeepSets | 51.26 | 0.32 | 0.35 | 0.37 | 9.10 | 80.99 | 62.85 | 22.90 | 20.96 |
| | | Transformer | 51.19 | 0.21 | 0.34 | 0.32 | 1.38 | 81.31 | 75.19 | 42.96 | 30.80 |

Table 10: **Variable-Dim Posterior Prediction:** Experimental results for posterior inference on variable dimensional datasets evaluated on estimating the (a) mean of a Gaussian distribution, (b) means of Gaussian mixture model (GMM), (c) parameters for linear regression (LR), and (d) parameters for linear classification (LC) for additional probabilistic model setups (eg. multi-class). We consider different backbone architectures and parametric distributions $q_\varphi$, and use dataset-specific bayesian and point estimates as baselines. $L_2$ Loss and Accuracy refer to the expected posterior-predictive $L_2$ loss and accuracy respectively. Here, cl refers to the number of clusters for GMM and number of classes for LC.

| Setup | | $q_\varphi$ | Model | $L_2$ Loss (↓) NLR | | | Accuracy (↑) NLC | | | | | |
|---|---|---|---|---|---|---|---|---|---|---|---|---|
| | | | | 1D | 50D | 100D | 2D-2cl | 2D-5cl | 50D-2cl | 50D-5cl | 100D-2cl | 100D-5cl |
| TANH | Baseline | - | Prior | 28.54 | 51.32 | 54.95 | 50.71 | 19.50 | 49.73 | 19.93 | 50.03 | 20.10 |
| | | - | Optimization | 0.27 | 17.54 | 33.97 | 97.86 | 93.66 | 69.55 | 42.35 | 65.11 | 35.33 |
| | | - | MCMC | 0.28 | 17.86 | 28.35 | 97.18 | 89.85 | 57.19 | 26.21 | 54.69 | 23.38 |
| | Fwd-KL | Gaussian | DeepSets | 27.93 | 51.44 | 55.01 | 49.35 | 20.32 | 50.06 | 19.91 | 49.96 | 19.91 |
| | | | Transformer | 27.43 | 51.08 | 55.35 | 49.34 | 20.52 | 50.07 | 20.04 | 49.95 | 20.06 |
| | Rev-KL | | DeepSets | 0.50 | 15.37 | 31.96 | 92.10 | 77.10 | 50.09 | 19.92 | 49.97 | 19.95 |
| | | | Transformer | 0.43 | 13.82 | 24.65 | 92.31 | 78.16 | 66.26 | 19.93 | 57.94 | 19.95 |
| | Fwd-KL | Flow | DeepSets | 31.23 | 49.49 | 56.85 | 49.15 | 20.70 | 50.17 | 19.78 | 50.53 | 20.30 |
| | | | Transformer | 30.87 | 48.49 | 57.23 | - | - | - | 21.16 | 19.93 | 20.23 |
| | Rev-KL | | DeepSets | 0.43 | 20.20 | 30.61 | 90.45 | 71.88 | 49.94 | 19.74 | 50.08 | 19.85 |
| | | | Transformer | 0.43 | 11.69 | 32.95 | 92.31 | 78.60 | 63.59 | 20.37 | 54.20 | 20.00 |
| RELU | | | Prior | 41.39 | 550.24 | 1066.89 | 50.92 | 19.87 | 49.86 | 19.95 | 50.43 | 20.01 |
| | | | Optimization | 0.32 | 96.28 | 261.19 | 96.90 | 94.20 | 74.22 | 54.10 | 71.20 | 48.06 |
| | | | MCMC | N/A | 104.11 | 278.48 | 96.53 | 90.59 | 67.11 | 39.20 | 65.53 | 34.95 |
| | Fwd-KL | Gaussian | DeepSets | 29.89 | 464.79 | 900.85 | 59.40 | 19.75 | 59.32 | 19.82 | 60.76 | 19.72 |
| | | | Transformer | 29.92 | 453.54 | 907.33 | 60.36 | 30.93 | 59.38 | 30.33 | 60.86 | 30.52 |
| | Rev-KL | | DeepSets | 0.58 | 149.36 | 370.32 | 89.67 | 61.72 | 62.78 | 34.21 | 64.45 | 34.25 |
| | | | Transformer | 0.56 | 83.55 | 259.64 | 89.35 | 74.12 | 72.59 | 36.33 | 69.69 | 35.33 |
| | Fwd-KL | Flow | DeepSets | 24.03 | 379.99 | 739.44 | 60.24 | 32.99 | 60.59 | 28.29 | 60.60 | 26.02 |
| | | | Transformer | 20.87 | 367.81 | 734.97 | 60.93 | 33.85 | 60.83 | 29.25 | 60.77 | 27.11 |
| | Rev-KL | | DeepSets | 0.57 | 150.69 | 355.97 | 88.25 | 58.51 | 63.55 | 31.17 | 63.64 | 28.07 |
| | | | Transformer | 0.48 | 87.65 | 295.94 | 90.02 | 74.56 | 71.74 | 38.43 | 66.94 | 31.00 |

Table 11: **Variable-Dim Posterior Prediction:** Experimental results for posterior inference on variable dimensional datasets evaluated on estimating the parameters of nonlinear regression (NLR) and classification (NLC) setups, with 1 layered MLP with different activation functions in the probabilistic model. We also consider a multi-class classification setup. We consider different backbone architectures and parametric distributions $q_\varphi$, and use dataset-specific Bayesian and point estimates as baselines. $L_2$ Loss and Accuracy refer to the expected posterior-predictive $L_2$ loss and accuracy respectively. Here, cl refers to the number of classes.

| Setup | | $q_\varphi$ | Model | $L_2$ Loss ($\downarrow$) NLR | | | Accuracy ($\uparrow$) NLC | | | | | |
|---|---|---|---|---|---|---|---|---|---|---|---|---|
| | | | | 1D | 50D | 100D | 2D-2cl | 2D-5cl | 50D-2cl | 50D-5cl | 100D-2cl | 100D-5cl |
| TANH | Baseline | - | Prior | 47.50 | 53.92 | 53.77 | 50.25 | 20.32 | 50.03 | 19.86 | 50.03 | 20.10 |
| | | - | Optimization | 0.43 | 38.92 | 48.70 | 97.61 | 92.65 | 65.67 | 35.57 | 60.55 | 30.07 |
| | | - | MCMC | 0.45 | 39.74 | 49.78 | 93.92 | 68.67 | 50.53 | 20.79 | 50.04 | 20.66 |
| | Fwd-KL | Gaussian | DeepSets | 47.78 | 53.72 | 53.76 | 49.70 | 19.98 | 49.86 | 20.04 | 49.62 | 20.09 |
| | | | Transformer | 47.20 | 53.92 | 53.86 | 49.71 | 20.14 | 49.85 | 20.18 | 49.63 | 20.16 |
| | Rev-KL | | DeepSets | 6.69 | 26.27 | 26.74 | 49.68 | 19.99 | 49.84 | 20.06 | 49.65 | 20.06 |
| | | | Transformer | 1.36 | 21.35 | 34.09 | 87.37 | 19.95 | 49.82 | 20.05 | 49.66 | 20.12 |
| | Fwd-KL | Flow | DeepSets | 48.22 | 52.32 | 48.74 | 50.16 | 18.57 | 49.97 | 20.01 | 49.95 | 20.16 |
| | | | Transformer | 47.90 | 53.31 | 49.83 | 50.04 | 18.76 | 50.14 | 20.12 | 49.86 | 20.23 |
| | Rev-KL | | DeepSets | 7.53 | 25.45 | 23.90 | 51.46 | 19.28 | 50.03 | 19.83 | 49.72 | 20.10 |
| | | | Transformer | 0.97 | 25.44 | 28.74 | 80.55 | 19.13 | 49.96 | 20.10 | 49.85 | 20.13 |
| RELU | Baseline | - | Prior | 670.13 | 9152.76 | 17988.61 | 49.58 | 20.50 | 50.23 | 19.76 | 49.95 | 20.56 |
| | | - | Optimization | 2.49 | 1557.89 | 4140.41 | 97.55 | 96.69 | 77.68 | 56.56 | 77.48 | 56.86 |
| | | - | MCMC | N/A | N/A | N/A | 64.63 | 25.76 | 62.28 | 28.31 | 62.73 | 30.82 |
| | Fwd-KL | Gaussian | DeepSets | 507.84 | 6989.40 | 13575.78 | 60.63 | 20.01 | 59.46 | 20.33 | 60.39 | 20.14 |
| | | | Transformer | 504.99 | 6921.67 | 13463.19 | 60.27 | 30.95 | 59.30 | 30.09 | 60.18 | 31.38 |
| | Rev-KL | | DeepSets | 5.93 | 2093.88 | 4508.67 | 76.89 | 54.92 | 67.21 | 45.37 | 68.75 | 49.43 |
| | | | Transformer | 4.29 | 1509.15 | 4128.72 | 82.55 | 58.95 | 67.30 | 45.11 | 68.58 | 48.18 |
| | Fwd-KL | Flow | DeepSets | 633.54 | 6280.19 | 10687.31 | 50.10 | 20.68 | 52.09 | 19.16 | 50.87 | 20.96 |
| | | | Transformer | 625.52 | 5378.48 | 9447.70 | 65.99 | 33.93 | 62.97 | 38.40 | 63.40 | 35.53 |
| | Rev-KL | | DeepSets | 4.12 | 2046.27 | 4151.37 | 82.87 | 60.21 | 70.87 | 60.04 | 72.22 | 51.70 |
| | | | Transformer | 1.78 | 1413.80 | 3539.80 | 90.75 | 64.79 | 70.88 | 59.71 | 73.27 | 50.00 |

Table 12: **Variable-Dim Posterior Prediction:** Experimental results for posterior inference on variable dimensional datasets evaluated on estimating the parameters of nonlinear regression (NLR) and classification (NLC) setups, with 2 layered MLP with different activation functions in the probabilistic model. We also consider a multi-class classification setup. We consider different backbone architectures and parametric distributions $q_\varphi$, and use dataset-specific Bayesian and point estimates as baselines. $L_2$ Loss and Accuracy refer to the expected posterior-predictive $L_2$ loss and accuracy respectively. Here, cl refers to the number of classes.

| | $q_\varphi$ | Model | LR | | | NLR | | | GP | | |
|---|---|---|---|---|---|---|---|---|---|---|---|
| | | | LR | NLR | GP | LR | NLR | GP | LR | NLR | GP |
| Baseline | - | Random | 3.000 | 18.4 | 1.955 | 3.000 | 18.47 | 1.955 | 3.000 | 18.4 | 1.955 |
| | - | Optimization | 0.242 | 0.74 | 0.053 | 0.242 | 0.741 | 0.053 | 0.242 | 0.74 | 0.053 |
| | - | MCMC | 0.247 | 3.64 | 0.062 | 0.247 | 3.643 | 0.062 | 0.247 | 3.64 | 0.062 |
| Fwd-KL | Gaussian | DeepSets | 0.248 | 0.70 | 0.059 | - | - | - | - | - | - |
| | | Transformer | 0.248 | 3.79 | 0.060 | - | - | - | - | - | - |
| Rev-KL | | DeepSets | 0.250 | 0.68 | 0.059 | 0.248 | 0.636 | 0.061 | 0.247 | 0.91 | 0.060 |
| | | Transformer | 0.249 | 2.35 | 0.061 | 0.246 | 0.637 | 0.060 | 0.250 | 5.65 | 0.061 |
| Fwd-KL | Flow | DeepSets | 0.247 | 1.31 | 0.060 | - | - | - | - | - | - |
| | | Transformer | 0.247 | 3.30 | 0.059 | - | - | - | - | - | - |
| Rev-KL | | DeepSets | 0.248 | 0.81 | 0.059 | 0.249 | 0.637 | 0.059 | 0.248 | 0.99 | 0.060 |
| | | Transformer | 0.246 | 1.72 | 0.058 | 0.245 | 0.641 | 0.058 | 0.246 | 4.53 | 0.059 |

Table 13: LR Model: Posterior predictive performance with L2 loss metric for the linear regression model. The top row highlights the data used to train the model (LR: Linear Regression, NLR: Nonlinear Regression (TANH), GP: Gaussian Process Regression), and the second row highlights the data used for evaluation. We note that a forward KL method can only be trained on data simulated from the assumed probabilistic model and thus cannot be trained on nonlinear data if the assumed probabilistic model is linear.

| Objective | $q_\varphi$ | Model | LR | | | NLR | | | GP | | |
|---|---|---|---|---|---|---|---|---|---|---|---|
| | | | *LR* | *NLR* | *GP* | *LR* | *NLR* | *GP* | *LR* | *NLR* | *GP* |
| Baseline | - | Random | 16.3 | 31.2 | 13.6 | 16.3 | 31.2 | 13.69 | 16.3 | 31.2 | 13.69 |
| | - | Optimization | 0.24 | 0.28 | 0.00 | 0.24 | 0.28 | 0.000 | 0.24 | 0.28 | 0.000 |
| | - | MCMC | 0.26 | 0.30 | 0.01 | 0.26 | 0.30 | 0.019 | 0.26 | 0.30 | 0.019 |
| Fwd-KL | Gaussian | DeepSets | - | - | - | 14.7 | 32.4 | 14.19 | - | - | - |
| | | Transformer | - | - | - | 14.3 | 32.0 | 13.90 | - | - | - |
| Rev-KL | | DeepSets | 0.33 | 0.76 | 0.14 | 0.34 | 0.40 | 0.112 | 0.35 | 1.05 | 0.115 |
| | | Transformer | 0.33 | 1.35 | 0.13 | 0.34 | 0.41 | 0.128 | 0.41 | 2.63 | 0.099 |
| Fwd-KL | Flow | DeepSets | - | - | - | 11.9 | 29.2 | 13.50 | - | - | - |
| | | Transformer | - | - | - | 12.5 | 13.2 | 12.38 | - | - | - |
| Rev-KL | | DeepSets | 0.31 | 0.74 | 0.11 | 0.31 | 0.38 | 0.080 | 0.32 | 0.84 | 0.081 |
| | | Transformer | 0.32 | 1.13 | 0.12 | 0.32 | 0.37 | 0.087 | 0.36 | 1.16 | 0.080 |

Table 14: NLR (TANH) Model: Posterior predictive performance with L2 loss metric for the nonlinear regression model with tanh activation function. The top row highlights the data used to train the model (LR: Linear Regression, NLR: Nonlinear Regression (TANH), GP: Gaussian Process Regression), and the second row highlights the data used for evaluation. We note that a forward KL method can only be trained on data simulated from the assumed probabilistic model and thus cannot be trained on linear or GP data if the assumed probabilistic model is a single-layered nonlinear MLP.

| Objective | $q_\varphi$ | Model | LR | | | NLR | | | GP | | |
| | | | LR | NLR | GP | LR | NLR | GP | LR | NLR | GP |
|---|---|---|---|---|---|---|---|---|---|---|---|
| Baseline | - | Random | 22.7 | 49.3 | 21.0 | 22.72 | 49.33 | 21.08 | 22.72 | 49.3 | 21.08 |
| | - | Optimization | 0.25 | 0.29 | 0.00 | 0.256 | 0.296 | 0.003 | 0.25 | 0.29 | 0.003 |
| | - | MCMC | N/A | N/A | N/A | N/A | N/A | N/A | N/A | N/A | N/A |
| Fwd-KL | Gaussian | DeepSets | - | - | - | 18.00 | 34.92 | 16.00 | - | - | - |
| | | Transformer | - | - | - | 17.22 | 34.08 | 15.30 | - | - | - |
| Rev-KL | | DeepSets | 0.28 | 3.16 | 0.10 | 0.310 | 0.381 | 0.074 | 0.302 | 1.42 | 0.069 |
| | | Transformer | 0.29 | 4.29 | 0.10 | 0.296 | 0.361 | 0.066 | 0.385 | 4.57 | 0.073 |
| Fwd-KL | Flow | DeepSets | - | - | - | 7.296 | 11.47 | 8.105 | - | - | - |
| | | Transformer | - | - | - | 9.863 | 12.53 | 10.34 | - | - | - |
| Rev-KL | | DeepSets | 0.27 | 0.85 | 0.09 | 0.290 | 0.351 | 0.059 | 0.288 | 3.84 | 0.059 |
| | | Transformer | 0.28 | 5.73 | 0.08 | 0.296 | 0.352 | 0.065 | 0.397 | 16.0 | 0.051 |

Table 15: NLR (RELU) model: Posterior predictive performance with L2 loss metric for the nonlinear regression model with ReLU activation function. The top row highlights the data used to train the model (LR: Linear Regression, NLR: Nonlinear Regression (RELU), GP: Gaussian Process Regression), and the second row highlights the data used for evaluation. We note that a forward KL method can only be trained on data simulated from the assumed probabilistic model and thus cannot be trained on linear or GP data if the assumed probabilistic model is a single-layered nonlinear MLP.

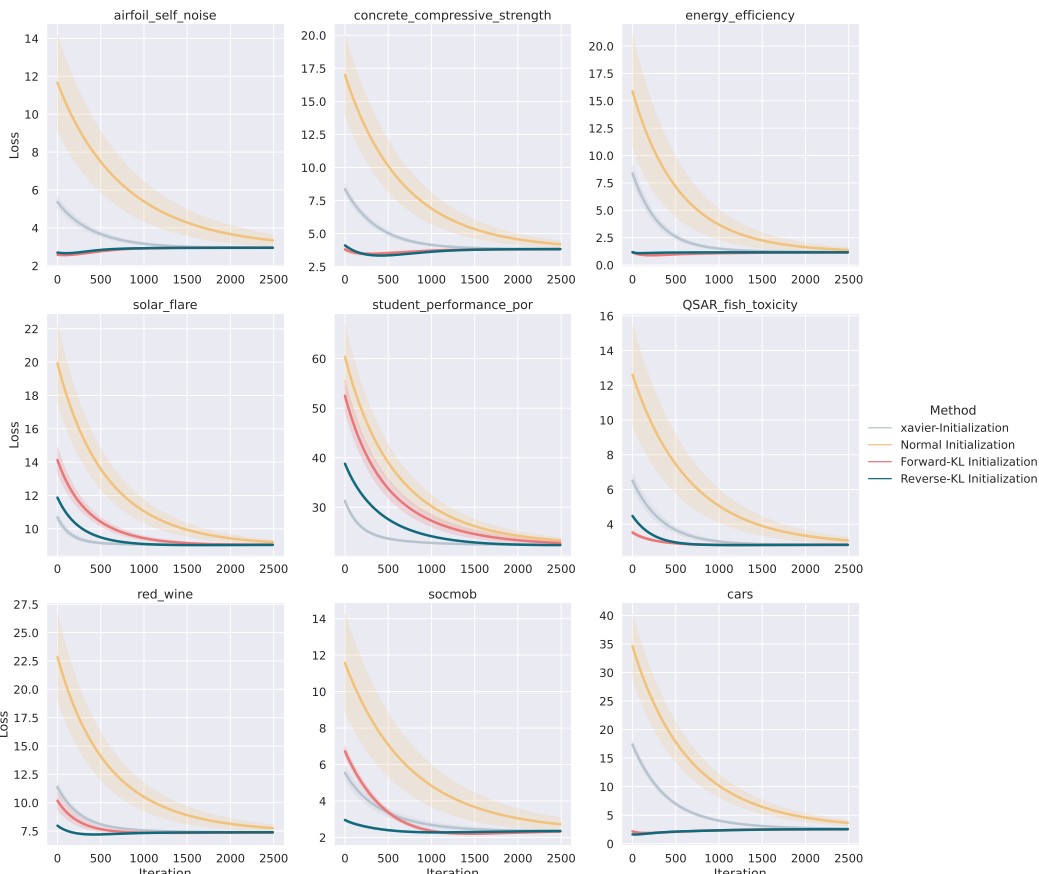

Figure 8: **Tabular Experiments | Linear Regression with Diagonal Gaussian**: For every regression dataset from the OpenML platform considered, we initialize the parameters of a linear regression-based probabilistic model with the amortized inference models which were trained with a diagonal Gaussian assumption. The parameters are then further trained with maximum-a-posteriori (MAP) estimate with gradient descent. Reverse and Forward KL denote initialization with the correspondingly trained amortized model. Optimization refers to a MAP-based optimization baseline initialized from the prior $\mathcal{N}(0, I)$, whereas Xavier-Optimization refers to initialization from the Xavier initialization scheme.

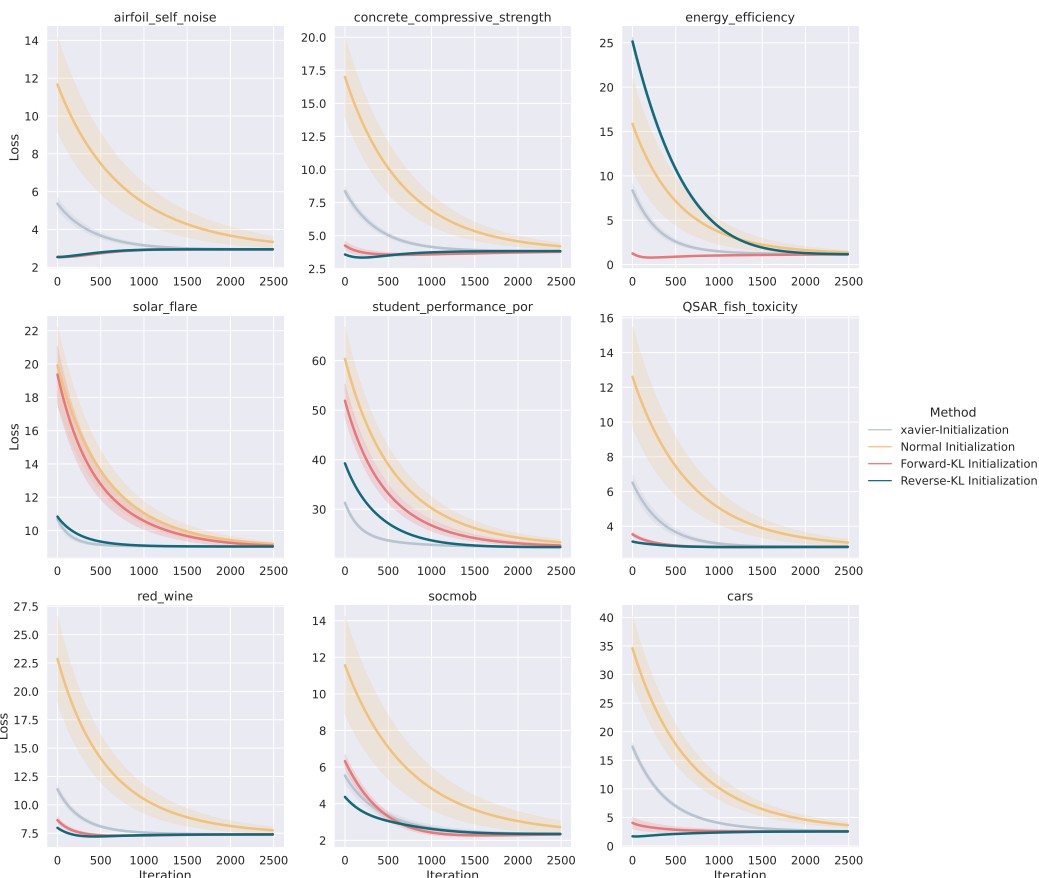

Figure 9: **Tabular Experiments | Linear Regression with Normalizing Flow**: For every regression dataset from the OpenML platform considered, we initialize the parameters of a linear regression-based probabilistic model with the amortized inference models which were trained with a normalizing flow-based model. The parameters are then further trained with maximum-a-posteriori (MAP) estimate with gradient descent. Reverse and Forward KL denote initialization with the correspondingly trained amortized model. Optimization refers to a MAP-based optimization baseline initialized from the prior $\mathcal{N}(0, I)$, whereas Xavier-Optimization refers to initialization from the Xavier initialization scheme.

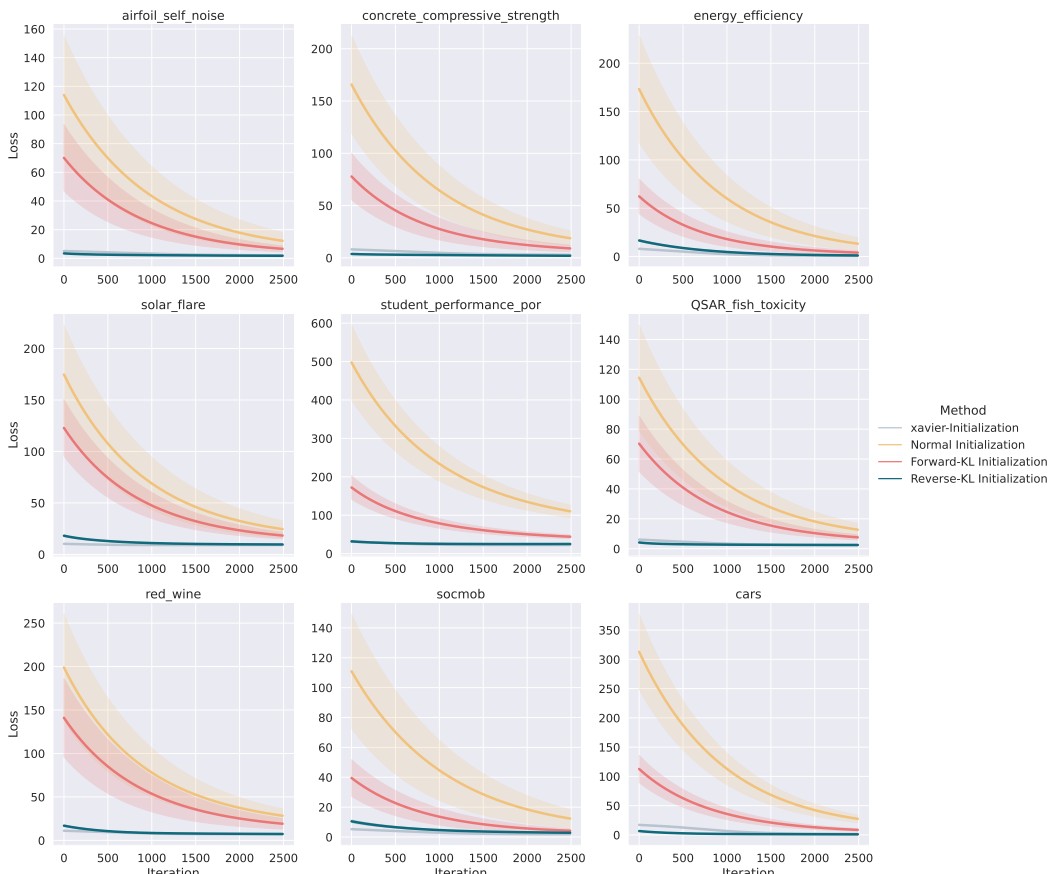

Figure 10: **Tabular Experiments | Nonlinear Regression with Diagonal Gaussian**: For every regression dataset from the OpenML platform considered, we initialize the parameters of a nonlinear regression-based probabilistic model with the amortized inference models which were trained with a diagonal Gaussian assumption. The parameters are then further trained with maximum-a-posteriori (MAP) estimate with gradient descent. Reverse and Forward KL denote initialization with the correspondingly trained amortized model. Optimization refers to a MAP-based optimization baseline initialized from the prior $\mathcal{N}(0, I)$, whereas Xavier-Optimization refers to initialization from the Xavier initialization scheme.

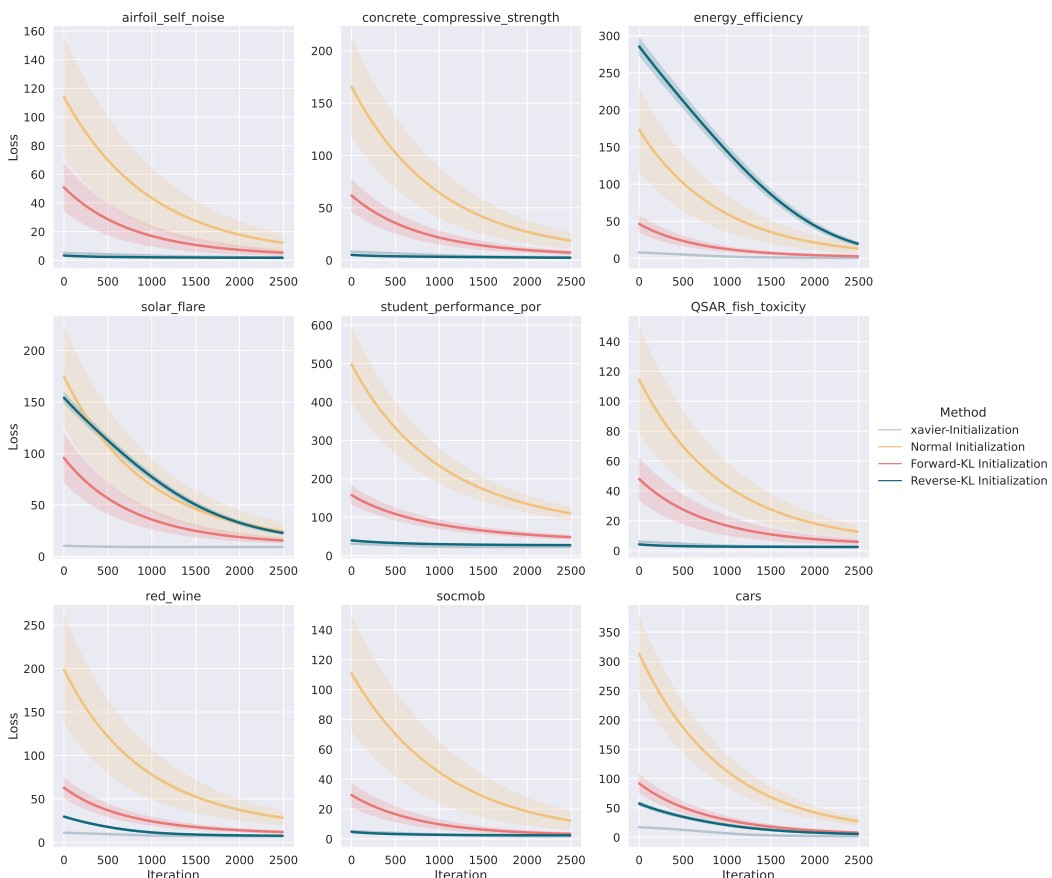

Figure 11: **Tabular Experiments | Nonlinear Regression with Normalizing Flow**: For every regression dataset from the OpenML platform considered, we initialize the parameters of a nonlinear regression-based probabilistic model with the amortized inference models which were trained with a normalizing flow-based model. The parameters are then further trained with maximum-a-posteriori (MAP) estimate with gradient descent. Reverse and Forward KL denote initialization with the correspondingly trained amortized model. Optimization refers to a MAP-based optimization baseline initialized from the prior $\mathcal{N}(0, I)$, whereas Xavier-Optimization refers to initialization from the Xavier initialization scheme.

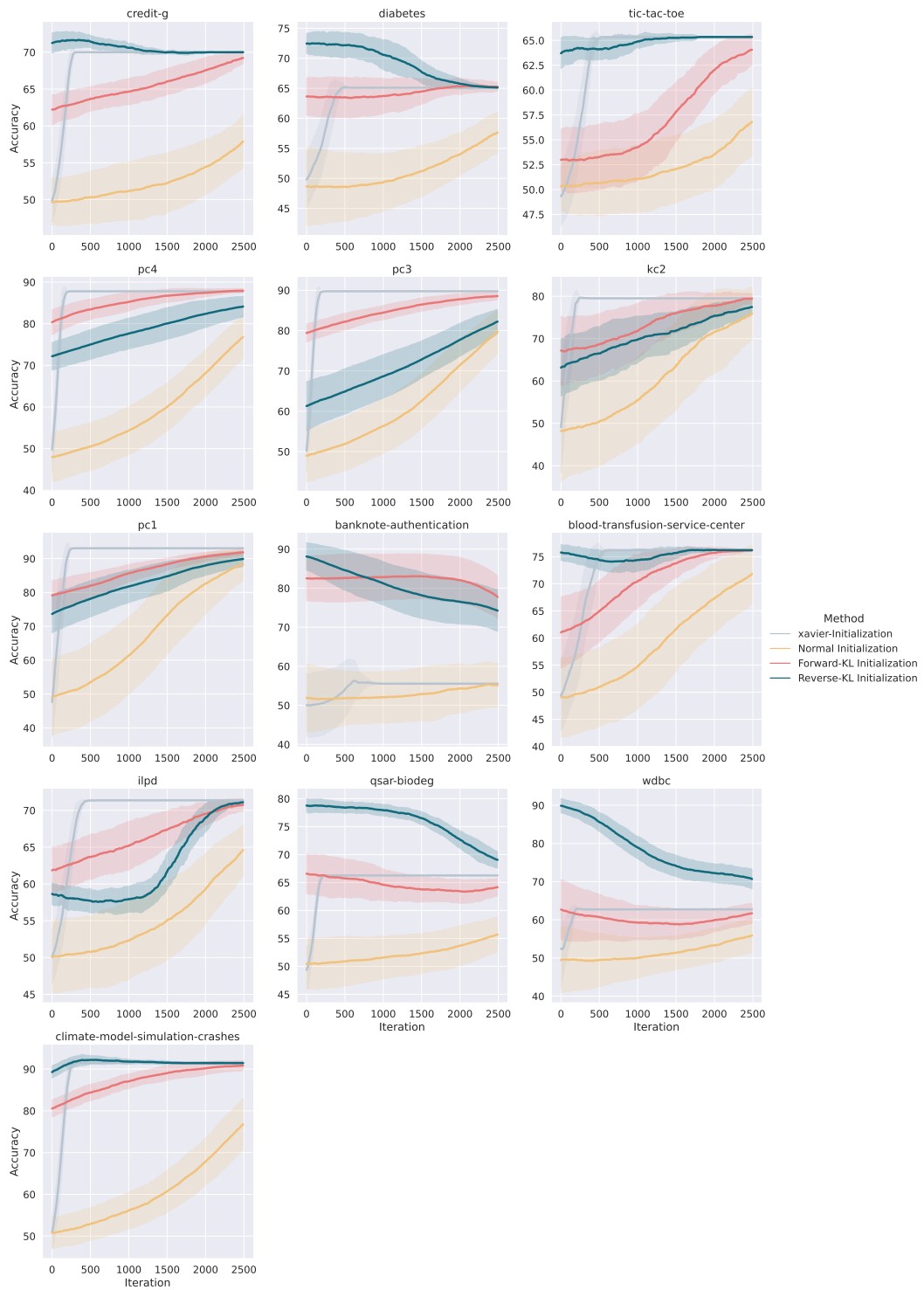

Figure 12: **Tabular Experiments | Linear Classification with Diagonal Gaussian**: For every classification dataset from the OpenML platform considered, we initialize the parameters of a linear classification-based probabilistic model with the amortized inference models which were trained with a diagonal Gaussian assumption. The parameters are then further trained with maximum-a-posteriori (MAP) estimate with gradient descent. Reverse and Forward KL denote initialization with the correspondingly trained amortized model. Optimization refers to a MAP-based optimization baseline initialized from the prior $\mathcal{N}(0, I)$, whereas Xavier-Optimization refers to initialization from the Xavier initialization scheme.

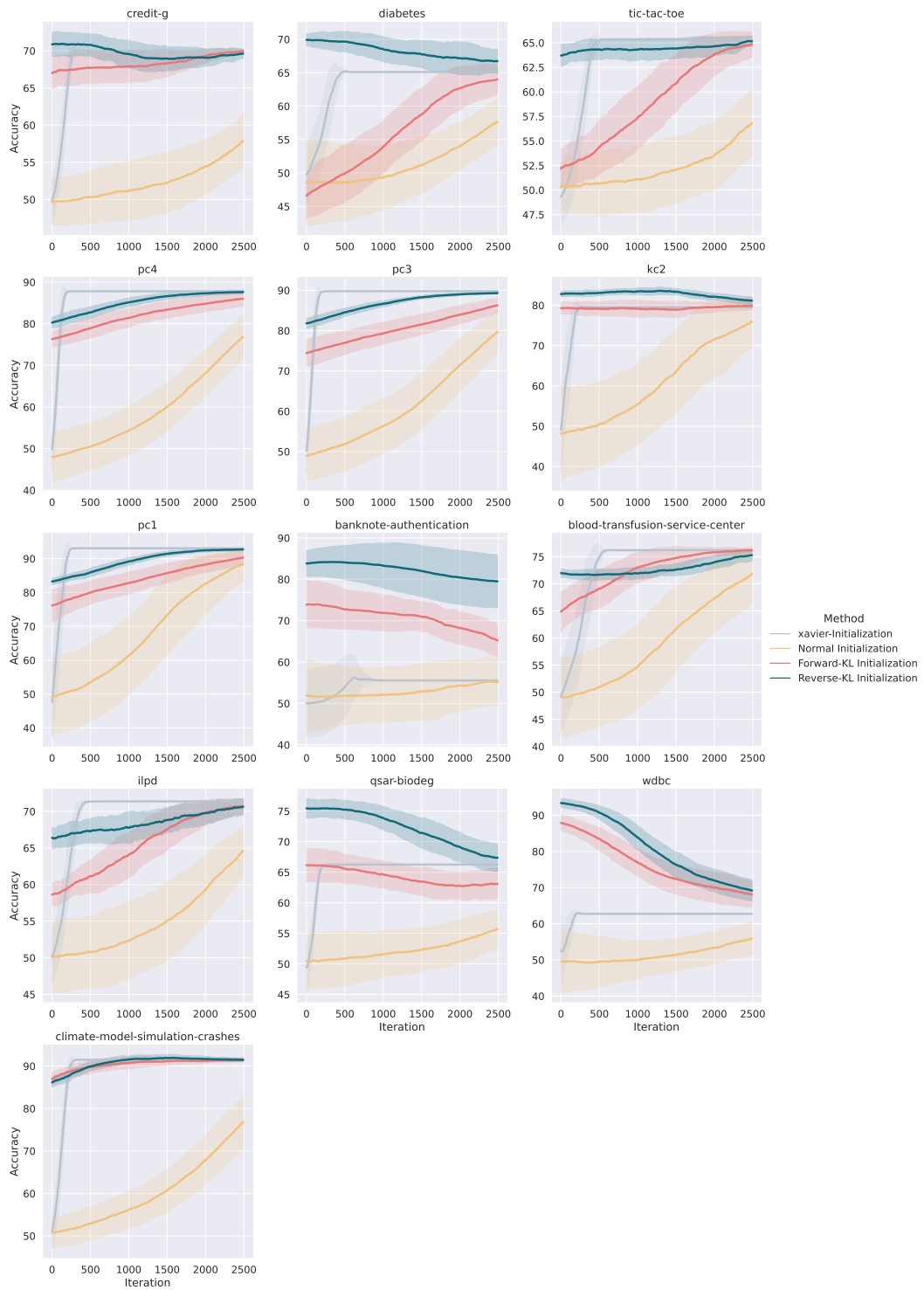

Figure 13: **Tabular Experiments | Linear Classification with Normalizing Flow**: For every classification dataset from the OpenML platform considered, we initialize the parameters of a linear classification-based probabilistic model with the amortized inference models which were trained with a normalizing flow-based model. The parameters are then further trained with maximum-a-posteriori (MAP) estimate with gradient descent. Reverse and Forward KL denote initialization with the correspondingly trained amortized model. Optimization refers to a MAP-based optimization baseline initialized from the prior $\mathcal{N}(0, I)$, whereas Xavier-Optimization refers to initialization from the Xavier initialization scheme.

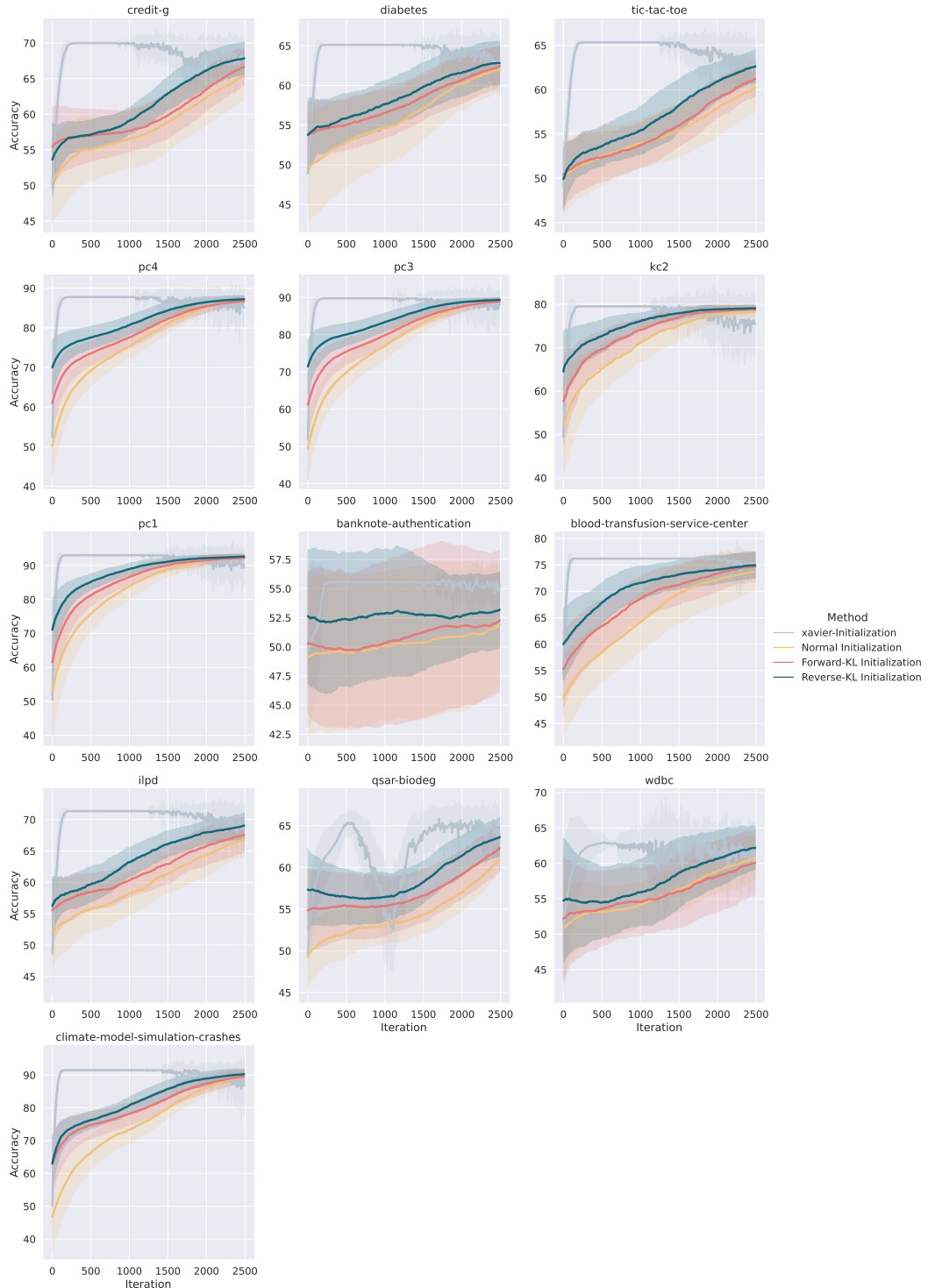

Figure 14: **Tabular Experiments | Nonlinear Classification with Diagonal Gaussian**: For every classification dataset from the OpenML platform considered, we initialize the parameters of a nonlinear classification-based probabilistic model with the amortized inference models which were trained with a diagonal Gaussian assumption. The parameters are then further trained with maximum-a-posteriori (MAP) estimate with gradient descent. Reverse and Forward KL denote initialization with the correspondingly trained amortized model. Optimization refers to a MAP-based optimization baseline initialized from the prior $\mathcal{N}(0, I)$, whereas Xavier-Optimization refers to initialization from the Xavier initialization scheme.

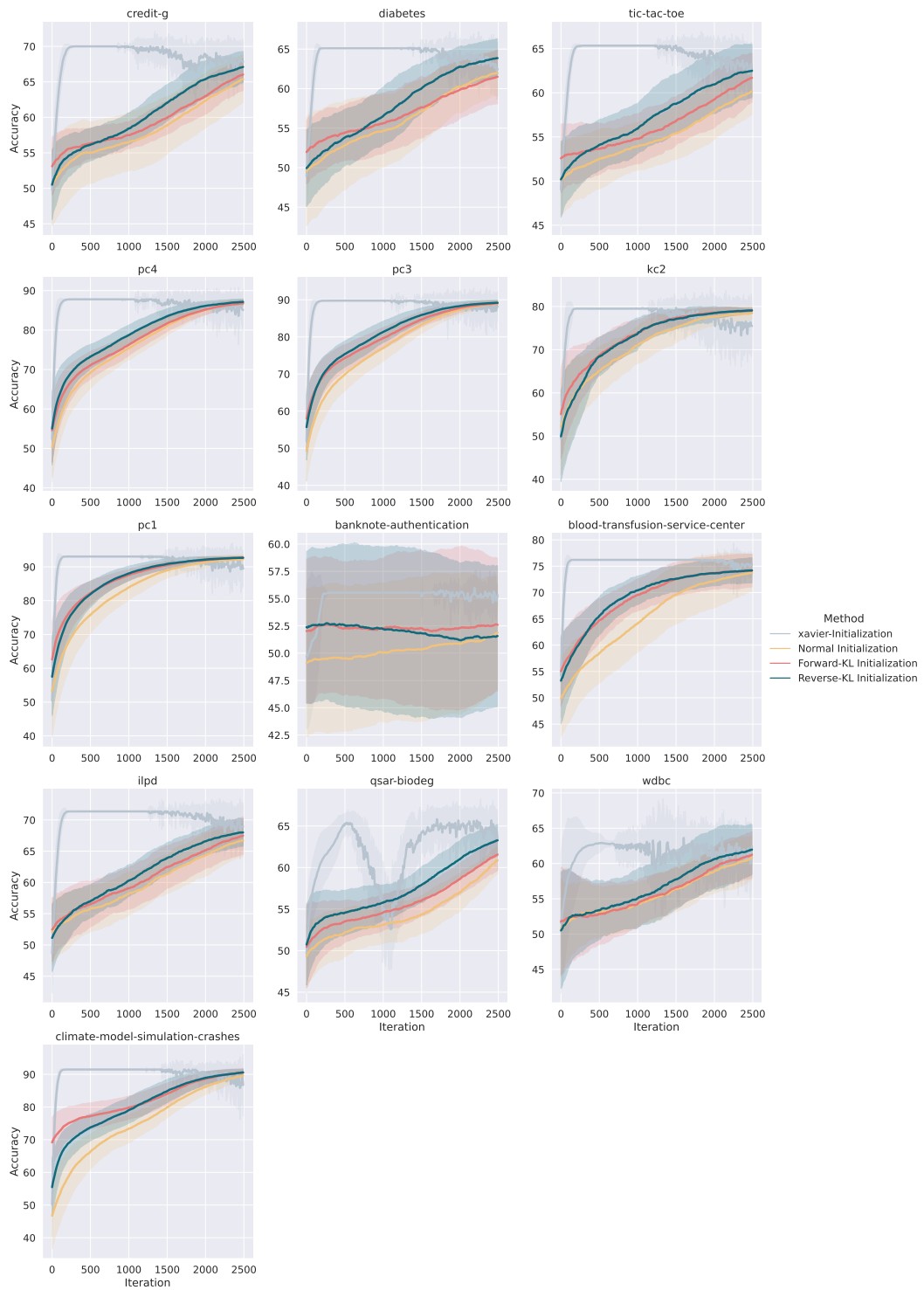

Figure 15: **Tabular Experiments** | **Linear Classification with Normalizing Flow**: For every classification dataset from the OpenML platform considered, we initialize the parameters of a linear classification-based probabilistic model with the amortized inference models which were trained with a normalizing flow-based model. The parameters are then further trained with maximum-a-posteriori (MAP) estimate with gradient descent. Reverse and Forward KL denote initialization with the correspondingly trained amortized model. Optimization refers to a MAP-based optimization baseline initialized from the prior $\mathcal{N}(0, I)$, whereas Xavier-Optimization refers to initialization from the Xavier initialization scheme.

