# OpenReview forum: "Exchangeable Dataset Amortization for Bayesian Posterior Inference"
_ICLR.cc/2024/Conference — Submitted to ICLR 2024_

### Official Review · Reviewer_xVE8 · 2023-10-19

**Soundness:** 2 fair
**Presentation:** 3 good
**Contribution:** 2 fair
**Rating:** 3
**Confidence:** 4

**Summary:**

The paper studies amortization schemes for variational Bayesian approximate inference methods. The approach is akin to the inference procedure used in standard variational autoencoders (maximize the reverse KL/ELBO), but amortization is performed over datasets instead of over individual datapoints. To solve this, the paper employs standard exchangeable aggregation architectures like deep sets or transformers.

In the experimental evaluation the authors consider a range of probabilistic models and explore two architectural choices: (i) deep sets vs. transformer-based aggregation (ii) Gaussian vs. normalizing flow variational posterior approximations. They compare against non-amortized inference schemes (max. likelihood, MCMC, random) as well as against an amortized approach based on the forward KL. The authors also evaluate how the methods perform under distribution shift.

**Strengths:**

The paper is mostly well written and the authors provide an extensive experimental evaluation with enough details to ensure reproducibility. I think that the architectural comparisons (Gaussian vs. normalizing flow/deep set vs. transformer) are interesting. Unfortunately, the experiments in their current form due not yet convince me of the paper's significance (see weaknesses).

**Weaknesses:**

My main concerns are (i) that the approach lacks novelty and (ii) that the experimental evaluation should be improved in various aspects.

Details:

(i) Amortized inference has been studied extensively in the past, e.g., in the context of the variational autoencoder. Amortization on the dataset level is also not new: it has been studied for years in the meta-learning community. In fact, the method is conceptually very similar to neural process (NP) [1] like approaches. The difference is that inference is performed over the decoder parameters directly and that, consequently, there are no free parameters that are optimized for predictive performance. While this is an architectural difference, it does not require any adaptations wrt to the posterior inference method (which is the only methodological proposal of the paper): both methods just optimize the ELBO wrt the variational parameters. The authors acknowledge these similarities, but argue that their method is new in the sense that NP-like methods are "predominantly designed for predictive modeling and thus cannot be used to provide useful information and uncertainty about model parameters". Unfortunately, the authors also largely focus on posterior predictive performance. The only results studying the quality of the approximate posterior are in Tab. 4 and Fig. 4 (c,d) which lack any comparisons against non-amortized baselines. I encourage the authors to explore further in which sense their method yields "useful information and uncertainty about model parameters", at least by adding more baselines to Tab. 4 (in particular baselines that allow to judge the amortization gap introduced by their method, cf. below).

(ii) Following my remarks above, I consider the paper's contribution to be exclusively empirical. While the provided architectural comparisons are interesting, I do not yet consider the contribution significant enough  to be of interest for the community. Thus, I encourage the authors to improve/extend their experimental evaluation:
- Please provide details about how the L2 and accuracy metrics were computed. (Please provide the exact formulae).
- Could you elaborate on why you do not evaluate the predictive log marginal likelihood instead of the L2 loss (as is typically done for assessing the predictive performance of Bayesian models). This metric should better measure the quality of predictive epistemic uncertainty estimates and, thus, implicitly of the posterior approximation.
- How much variance is introduced in the results due to the algorithm's/network's initialization? Please provide confidence intervals for your experimental results. In its current form it is impossible to judge their significance.
- I would propose to also add a non-amortized version of the proposed method as a baseline. This would allow to judge the amortization gap, i.e., the approximation error introduced by amortization alone.
- The authors state that normalizing flows do not increase approximation accuracy due to the mode-seeking behavior of the reverse KL objective. I would be interested in a discussion and/or comparison to recent natural-gradient based methods such as [2,3] that perform VI with expressive Gaussian mixture approximations by inducing terms in the objective that prevent mode collapse.
- The authors argue that deep sets are inferior to transformer-based architectures because of the naive sum/mean-based aggregation of deep sets. [4] propose a Bayesian aggregation method that tackles exactly this problem. It would be interesting to see how Bayesian aggregation compares to transformer-based aggregation.

[1] https://arxiv.org/abs/1807.01622
[2] https://arxiv.org/abs/2002.10060
[3] https://openreview.net/pdf?id=tLBjsX4tjs
[4] https://openreview.net/forum?id=ufZN2-aehFa

**Questions:**

See my comments below "Weaknesses".

**Details Of Ethics Concerns:**

---

---

> ### Author Response · Authors · 2023-11-16
> **Author Response**
>
> We thank the reviewer for their valuable comments. Our response aims to address the raised concerns, and we are available for further discussions to resolve any outstanding comments or ambiguities.
>
> **Contribution and Metrics**
>
> We refer the reviewer to the common shared response which provides a discussion about the novelty of our work as well as the different contributions made. It also refers to the exact formulae used to define the metrics that we consider.
>
> **Novelty**
>
> We would like to respectfully disagree with the reviewer on their point about novelty. While we agree that the methodologies used (eg. Variational Inference (VI), amortization on the dataset, transformers) do exist in current literature, it is important to note that its application for amortized Bayesian posterior estimation hadn’t been explored yet and this is precisely the void that we aimed to fill. It is true that in hindsight it might look like a simple application of existing concepts but this gap does indeed exist in literature and we feel that there is merit and contribution in addressing it since the class of parametric models with tractable likelihoods do span a significant volume of models used by practitioners. Thus, leveraging and combining existing methods to allow for faster and more efficient posterior estimation in such a class of models is important.
>
> Additionally, as the reviewer points out, Neural Processes (NPs) are only applicable in predictive tasks and thus our proposed approach does tackle a different problem. In particular, our objective is not the same as NPs as we optimize the ELBO with respect to the variational parameters only (there are no other parameters in the likelihood model) whereas NPs optimize the ELBO with respect to both variational parameters as well as the parameters in the likelihood model. Though seemingly similar, it is important to note that the former is a direct application of the VI framework while the latter of Variational Expectation Maximization.
>
> Finally, we would also like to point out that neither NPs nor Simulation-based Inference (SBI) approaches generally handle variable dimensional inputs in modeling the predictive tasks. Further, the two directions have mostly seen progress separately, without works comparing them under the common setting of probabilistic models with tractable likelihoods. Additionally, we also consider model mis-specification and OoD generalization based real world settings which have been relatively less explored in both SBIs and NPs.
>
> Predictive log marginal likelihood: We are not quite clear what the reviewer means by this quantity, but we do add another metric definition in Appendix D (CNLL). An example of this metric in our proposed experiments is available in Tables 5 and 6. The reason we did not include this metric was because it was fairly well correlated with downstream $L_2$ or accuracy metrics, but for completeness we are happy to include it in the Appendix. We request the reviewer to let us know if they had a different metric in mind.
>
> **Related Work**
>
> We thank the reviewer for pointing out some of the relevant works on VI approaches leveraging a Gaussian Mixture distribution as the approximating distribution as well as different context averaging methods like Bayesian Context Averaging. We believe that empirical comparisons against these approaches is a contribution on its own and thus leave it as future work. However, we do include a discussion on them in the main text as well as the Appendix in the revised draft.
>
> **Variance due to Initialization**
>
> We agree with the reviewer that it would be nice to have an estimate of variance due to initialization in the posterior estimation. We would like to point out, however, that all our metrics are evaluated over 100 datasets, each with 25 samples from the approximate variational distribution which already provides an estimate of the reliability of the model. We omit the standard deviation in this setup so as to improve readability in the tables, but we will open-source our code for the community to freely use the approach. The experiments are computationally extensive to run since we test on a wide variety of probabilistic models but we would be happy to attempt to provide some estimates. Is there a particular metric / setup that the reviewer is interested in seeing the effect of initialization on?
>
> **Non Amortized Baseline**
>
> We thank the reviewer for this point and note that we do consider two non amortized baselines, optimization and MCMC. These baselines in particular show that the amortized VI setup is not as strong as optimization in predictive modeling but does compare significantly favorably to MCMC. We also decided to opt out of KDE estimates of MCMC in Figure 4 just to make the figures clearer as we already have the true posterior on it.
>
> We hope that our response clarifies the reviewer's concerns and we would be happy to answer any additional questions and concerns.

---

> > ### Comment · Reviewer_xVE8 · 2023-11-22
> > **Thanks for the feedback**
> >
> > I thank the authors for their answer and clarifications. Unfortunately, I'm still not convinced that the manuscript is ready for publication at a conference like ICLR. I also feel like my view is in line with the other reviewers, so I'll keep my score. Please find some details for my decision below.
> >
> > - While I agree that the paper considers "pure variational inference" in contrast to NP's variational expectation maximization, I do not think that this fact alone counts as a relevant theoretical/methodological contribution. In fact, the theoretical discussion (Sec. 3) as well as architectural approaches (Sec. 4) are well known, i.p., in the meta-learning community. I also agree that NPs perform inference over a conceptually different set of variational parameters (i.e., not over the likelihood aka decoder parameters, but over the inputs to the decoder), but from an abstract point of view this does not require any changes to the methodology used. Moreover, "purely variational" approaches have also been studied in the meta-learning community for a long time, e.g., in probabilistic/Bayesian extensions of MAML such as [1]. Finally, I agree with the other reviewers that the author's masking-based approach to handling variable dimensionality is neither new nor theoretically especially appealing. Therefore, I'm still convinced that the paper's contribution is exclusively empirical.
> > - As mentioned in my initial review, I in principle *do* consider empirical papers valuable for the community (as noted, some of the architectural comparison provided in the paper are indeed interesting). Unfortunately, I do not think that the provided experimental evaluation of the author's manuscript overall provides enough benefit for the community to be ready for publication. I.p., I still do not see how the author's claimed contribution
> >
> > "Instead, our primary objective is to infer the posterior distribution over them. This nuanced differentiation in methodology contributes to the unique positioning of our approach within the broader landscape of Bayesian posterior estimation methodologies. [...] our contribution lies in their application to the specific challenge of Bayesian posterior estimation in probabilistic models with tractable likelihood functions."
> >
> > is reflected in the experimental evaluation. Concretely, I do not see the central question answered why performing "pure" variational inference in the likelihood's parameters space is beneficial in comparison to established approaches from the NP or MAML [1] families. To reiterate my concerns:
> > - Except for simple toy examples, the authors also only compare predictive performance, but do not include comparisons against the mentioned baselines from the meta-learning literature.
> > - I'm not convinced that the used predictive metrics (L2/accuracy) are suitable as proxies for Bayesian posterior estimation quality. More precisely: I think that *predictive* metrics can in principle be used as proxies for *posterior* estimation. However, I'm not convinced that predictive *L2/accuracy* is suitable (as noted by the authors, the added conditional NLL metric should be largely equivalent to predictive L2). How do these metrics measure the quality of epistemic uncertainty estimates? What I meant in my initial review is using the log marginal predictive likelihood, as for example used in [2], Eq. 16. I'd be interested in the author's view on the distinction between the metrics and why they chose L2 over log marginal predictive likelihood.
> > - While the authors include a discussion with MAML/NP like approaches, "the comparison is often limited to a few sentences that are, in my opinion, vague and insufficient" (as noted also by reviewer iJJB).
> >
> > [1] https://arxiv.org/pdf/1806.02817.pdf
> > [2] https://openreview.net/pdf?id=sb-IkS8DQw2

---

> > > ### Author Response · Authors · 2023-11-23
> > > **Author Response**
> > >
> > > We want to thank the reviewer for the fruitful discussion and ideas mentioned. In particular, how to frame the presented novel method in the picture of other purposed methods (GP/NP/Meta-Learning) that utilize amortized inference for posterior prediction and parallel research streams like SBI that differ in the problem setup.
> > >
> > > As mentioned by other reviewers, it is now brought to our attention that we need to focus on a more precise differentiation between existing methods and our proposed novel method. We want to thank the reviewer for highlighting methods from meta-learning and ideas for different aggregation schemes, which we will review carefully.
> > >
> > > Furthermore, we thank the reviewer for clarifying the log marginal posterior predictive metric. In our setup, the likelihood is not learnable and only parameterizable by the posterior parameters $\theta$ sampled from the approximate posterior distribution $\theta_s \sim q_\varphi(\theta | \mathcal{D})$ of the whole given dataset, e.g., $\mathcal{D} = \\{(y_n, x_n)\\}$.
> > > Further, since we do not use a context set for conditioning the marginal mentioned by the reviewer, this quantity would reduce to computing $\log p(y|x)$, which does not depend on $\theta$ and thus is constant irrespectively of forward or reverse KL.

---

### Official Review · Reviewer_iJJB · 2023-10-31

**Soundness:** 2 fair
**Presentation:** 1 poor
**Contribution:** 2 fair
**Rating:** 3
**Confidence:** 3

**Summary:**

This manuscript proposes a method for amortized Bayesian posterior inference. In particular, this method leverages set-based neural network architectures to design an amortized posterior that can deal with observation of varying cardinality. The model is trained using the reverse-KL divergence which is shown empirically to work better for the considered benchmarks. The authors also show that this leads to better performance when the model used differs from the data-generating process.

**Strengths:**

* The method is sound.
* Building amortized Bayesian inference algorithms that can deal with sets of observations of different cardinality and be robust to model specification is significant.

**Weaknesses:**

Overall, I find that the paper lacks of clarity making it hard to follow. Here is a list of things that, in my opinion, harms clarity:

* The contributions are not clear. Here are the three claimed contributions:
   1. *"Proposing a novel method for performing Bayesian inference in probabilistic models solely through inference on a trained amortization network, and demonstrating its effectiveness in a variety of settings and with several well-known probabilistic models."*
Throughout the paper, it is not clear to me what is claimed as novel in the proposed method. Is performing Bayesian inference based on a trained amortization network claimed to be novel? Is using the reverse KL divergence novel? Is the fact of using a backbone that accepts sets as input to handle datasets of different cardinality novel?

	2. *"Providing insights into various design choices like the architectural backbone used and the choice of parametric distribution through detailed ablation experiments."* Ok

	3. *"Highlighting the superior performance of our proposed approach when compared to existing baselines, especially in the presence of model misspecification and real-world data."* Does the contribution lie in the design of a new method to handle model misspecification or the empirical study of existing methods in this context?

* There are figures all over the place while they all belong to section 4. It is very confusing to see experimental figures in the middle of the introduction. In addition, when reading the experiment section, the reader has to jump back to the introduction to see the figure.

* Equation (8) seems to be very similar to equation (9) from prior work. Would it be easier to start from there, explain what $\chi$ is and say that you use the reverse KL instead? I feel that previous explanations dilute the message and make things hard to follow while equation (9) is straightforward to understand.

* Section 4.3 seems to be full of methodological elements while being in the middle of the experiments section. I think grouping all the methodological elements together would help clarify the contributions.

* In section 4.3 it is said that "In contrast, we can leverage our proposed reverse KL approach to train an amortized inference model to predict the posterior over the assumed probabilistic model’s parameters by directly using the available unpaired data during training." It is not clear to me how this is done while this seems to be a contribution of the paper. It would be worth to expand on this more.

I think the following paper should be discussed in the related works. It addresses the problem of amortized Bayesian inference for datasets of different cardinality. It exploits the fact that the scores of each individual observation can be composed to produce the score of the joint observations. This joint score can then be used to efficiently produce samples from the posterior distribution.
Geffner, T., Papamakarios, G., & Mnih, A. (2023). Compositional Score Modeling for Simulation-Based Inference.

In the experiments, the quality of the approximate posteriors is quantified using either the expected $L_2$ loss or the expected accuracy loss. This is unclear to me what those losses are. I think it is important to include their mathematical definition in the manuscript. From what I understood, the expected $L_2$ loss can be defined as
$L_2 = E_{p(\theta|D)}[(\theta - \tilde{\theta})^2] $
where $\tilde{\theta}$ would be the parameters used to generate $D$. I think this metric is unsuited when the posterior is multimodal. An approximation that puts the mass in the middle of the two modes (where there should be no mass) will have a lower $L_2$ than an approximation that puts half the mass in each mode.

I cannot assess the novelty due to the lack of clarity regarding the contributions.

**Questions:**

* Could you clarify what in the manuscript is a contribution and what belongs to the background?

* Could you clarify what are the quantities used for evaluation and justify the use of those?

---

> ### Author Response · Authors · 2023-11-16
> **Author Response**
>
> We sincerely appreciate the reviewer's valuable comments. In our response, we aim to address and clarify the majority of the concerns raised and would be happy to engage in additional discussions should there be any remaining unresolved comments.
>
> **Contribution and Metrics**
>
> We refer the reviewer to the common shared response which provides a discussion about the novelty of our work as well as the different contributions made. It also refers to the exact formulae used to define the metrics that we consider.
>
> **Starting from Equation 8-9**
>
> We argue that the reason Equation 9 seems familiar from prior work is because it is the general starting point of defining the optimization problem for Simulation-based Inference (SBI) approaches. However, we rely on the reverse KL / Variational Inference (VI) framework where the derivation leads to the well studied ELBO objective. Thus, we discuss the connections to ELBO and amortization more clearly in the start to provide a complete story as well as the exact loss formulation, which is defined in Equation 7. Mathematically, equations 7 and 8 describe an equivalent optimization problem.
>
> **Section 4.3**
>
> We provide each section of the experiments with its own set of methodological details and defer the readers to the Appendix for additional tangential details (eg. the choice of mis-specification explored). This is because we want each experimental section to be self-contained about the experiments discussed there. We will take the reviewer’s concerns into account and update Section 4.3 in the next revision to clarify some of the concerns regarding the setup.
>
> To clarify, we would like to point the reviewer to the differences in Equations 8 and 9. Equation 9 is only tractable for training if $\chi$ defines sampling according to the probabilistic model $p(\mathcal{D} | \mathbf{\theta})$, since then we observe the pairs $\{(\mathcal{D}, \mathbf{\theta})\}$. However, in general, we often only observe a stream of data $\{\mathcal{D}\}$ from some data generating function $\chi’$ coming from an unknown probabilistic model, without its corresponding parameters $\mathbf{\theta}$. In this setting of model mis-specification, we find that when we train both the proposed method and SBI baseline on $\chi$ and evaluate on $\chi’$, the proposed method generalizes better.
>
> Additionally, even though we don’t know the underlying probabilistic model, we can still model the observations relatively well. Practitioners often define a probabilistic model to explain this stream of data; which could be different from the ground-truth underlying probabilistic model and hence wrong. However, one could still model the data well by finding a good set of parameters of this ill-specified probabilistic model. A caveat of SBI approaches, which are generally framed as a Forward KL optimization problem, is that they cannot be trained using data obtained from $\chi’$ and thus would not be able to leverage diverse data coming from different and unknown probabilistic models. In contrast, our proposed method can leverage such data during training, and we show in Table 3 that leveraging such data improves performance even further.
>
> More succinctly, this difference arises from swapping $\chi$ with $\chi’$ in Equations 8 and 9 such that $\chi’$ does not follow $p(\mathcal{D} | \mathbf{\theta})$ while $\chi$ does. This swap makes Equation 9 infeasible to optimize, while Equation 8 still remains feasible. We hope that this clarifies the reviewer’s doubts regarding how this setting is formulated.
>
> As an example, we can consider the probabilistic model as a nonlinear Bayesian Neural Network model. It can only be trained under the SBI framework if $\chi$ represents nonlinear data sampled according to this model. However, we might receive a stream of data of interest that might follow a linear relationship. This data cannot be part of the training framework in the SBI setup, but can be in our proposed method.
>
> **Related Work**
>
> We have added a discussion about the work on score modeling for SBI in the Appendix. We were not aware of this work but as we point out, it is still different from the proposed approach since we are not leveraging the SBI framework for amortized posterior estimation and instead propose a complementary research stream and show benefits of performing amortized VI. However, we provide a small discussion on the similarities in the Appendix A of the revised draft.
>
> **Figure Positioning**
>
> We apologize for the figure positioning and will try to improve it where possible.
>
> We hope that our response clarifies the reviewer's concerns and we would be happy to answer any additional questions and concerns.

---

> > ### Comment · Reviewer_iJJB · 2023-11-20
> >
> > Thank you for this detailed response! I will here share my updated view on the different points raised both here and in the common response.
> >
> > First, I should say that many things have been clarified with this response. It shows the value of this work but also strenghten the need for substantial rewritting. It is now clear to me that, in its current form, the paper lacks too much of clarity to be accepted but that it can become valuable work if rewritten. It particular, I think the paper would greatly benefit from a clearer explanation of how this work is positionned in the litterature and what differs from existing work in SBI/NP/GP. The last paragraph of section 4.3 should also be extended to explain how such data can be leveraged as this seems to be a contribution of this work.
> >
> > I also have to point out this sentence with which I disagree.
> >
> > > Moreover, it is crucial to emphasize that to the best of our knowledge, none of the existing methods within SBI or NPs demonstrate the capability to effectively handle inputs of variable dimensions while our method can, which in itself is a novel contribution.
> >
> > I have to disagree because this is precisely what is done in Geffner, T., Papamakarios, G., & Mnih, A. (2023). Compositional Score Modeling for Simulation-Based Inference. It constructs a score model that can handle inputs of variable dimensions and be used to sample from the posterior.

---

> > > ### Author Response · Authors · 2023-11-20
> > > **Author Response**
> > >
> > > We thank the reviewer for their response. We would like to politely disagree with the reviewer as we do mention how we tackle a different problem than the NP framework; as well as we do perform a direct comparison with SBI, which is the Forward KL approach. Details of how we are different is also present in the last paragraph of Section 2. Additionally, we believe that GPs are only tangentially related to the work in so much as how amortization has been used to learn the kernel function for GPs. We additionally provide discussion with all such related work in Appendix A as well as describe how we are different in more detail.
> > >
> > > *"I have to disagree because this is precisely what is done in Geffner, T., Papamakarios, G., & Mnih, A. (2023). Compositional Score Modeling for Simulation-Based Inference. It constructs a score model that can handle inputs of variable dimensions and be used to sample from the posterior."*
> > >
> > > We think that maybe the reviewer is confusing by what we mean by dimensionality. Both our model and Geffner et. al can handle variable number of data-points / observations when modeling the posterior distribution. In our work, we provide an additional flexibility by allowing for modeling observations lying on variable dimensional spaces. That is, the same network can be used to model the posterior over weight vectors for 2-D regression as well as 5-D regression. Geffner et. al on the other hand don't look at this axis of variable dimensionality, and instead look at amortizing over variable number of data-points.
> > >
> > > Thank you again for your thoughtful review. We made a significant effort to address your feedback and include additional experiments, and would appreciate it if you would consider raising your score in light of our response. Please let us know if there are further questions we can address.

---

> > > > ### Comment · Reviewer_iJJB · 2023-11-22
> > > >
> > > > Thanks for your response.
> > > >
> > > > Other approaches are indeed mentioned but the comparison is often limited to a few sentences that are, in my opinion, vague and insufficient. Also, this is only a small example of a more general lack of clarity. Regarding the variable dimensionality, my thinking is that handling variable dimensional spaces is something more related to the neural network architecture used than the inference algorithm. Geffner et. al work could easily be extended to handle variable dimensional spaces by modeling the score with an appropriate neural network. For those reasons, I will leave my score unchanged.

---

> > > > > ### Author Response · Authors · 2023-11-23
> > > > > **Author Response**
> > > > >
> > > > > We want to thank the reviewer for the fruitful discussion and ideas mentioned. In particular, how to frame the presented novel method in the picture of other purposed methods (GP/NP) that utilize amortized inference for posterior prediction and parallel research streams like SBI that differ in the problem setup.
> > > > >
> > > > > We further agree that while a number of approaches can be extended to handle variable dimensionality, they currently do not, in light of which it is a contribution of our work.

---

### Official Review · Reviewer_oaRk · 2023-10-31

**Soundness:** 3 good
**Presentation:** 3 good
**Contribution:** 2 fair
**Rating:** 5
**Confidence:** 4

**Summary:**

The paper presents a framework for amortized inference of probabilistic model parameters, $\theta$, using neural networks that maintain permutation equivariance among data points. The neural network takes in a dataset of some number of datapoints and provide amortized inference of the model parameters, it is also capable of taking in datasets of various dimensions.

The method doesn’t require ground truth parameters during training. Instead, the objective is to minimize the reverse KL divergence under the Variational Inference (VI) framework, with the requirement that the model provides a closed-form likelihood of the data given the model parameters. The method is applied to probabilistic models such as (non-)linear regression/classification and Gaussian mixtures. On unseen problems, the amortized inference parameters serve as good initial guess that can speed up optimization based on that.

**Strengths:**

- *Versatility of Application*: The method is demonstrated to be effective for both fixed and variable-dimension parameter spaces. In the latter case, the method cleverly leverages masking to manage unused dimensions.
- *Model Robustness*: By adopting a reverse KL approach rather than forward KL, the presented model offers greater resilience to model-misspecification. This implies that it can effectively handle cases where training datasets and test datasets might be characterized by different underlying probabilistic models.

**Weaknesses:**

- *Literature Gap*: The paper seems to omit relevant literature on amortized GP hyperparameters [1][2][3]. And GP belongs to the framework considered here because it has closed form likelihoods given model parameters. Specifically,[1] produces amortized inference for point estimate of the posterior for GP hyperparameters given a dataset. The neural network architecture proposed in [1] also makes use of transformer for permutation equivariance. [1][2][3] also generalize to unseen datasets, with the same meta-learning flavor. This paper broadens the perspective by considering general probabilistic models and considering a distribution rather than a point estimate. But the basic idea and architecture choices share the same spirit, which decreases the novelty in the methodological contribution.
- *Variable Dimension Handling*: The method of managing variable-dimensions through masking might be limited. It wastes GPU memory and is not equivariant w.r.t. dimensions. There could be potential benefits in exploring the neural network employed by [1] if each dimension has its own parameters, such as in linear regression and GMM.
- *Clarity on GMM*: It is unclear how the proposed approach would manage variable  number of mixtures in the case of Gaussian Mixture Models (GMM).
- *Ablation Limitations*: While the mention of an ablation study is commendable, it would be beneficial to see a more comprehensive study, for instance, sweeping across dimensions ranging from 1-100D, to see how the approach extrapolates on dimensions different than the training data.

[1] Liu, Sulin, Xingyuan Sun, Peter J. Ramadge, and Ryan P. Adams. "Task-agnostic amortized inference of gaussian process hyperparameters." Advances in Neural Information Processing Systems 33 (2020): 21440-21452.

[2] Simpson, Fergus, Ian Davies, Vidhi Lalchand, Alessandro Vullo, Nicolas Durrande, and Carl Edward Rasmussen. "Kernel identification through transformers." Advances in Neural Information Processing Systems 34 (2021): 10483-10495.

[3] Bitzer, Matthias, Mona Meister, and Christoph Zimmer. "Amortized Inference for Gaussian Process Hyperparameters of Structured Kernels." UAI (2023).

**Questions:**

- *Choice of MCMC*: What was the motivation behind choosing Langevin instead of HMC? Further, is the paper referring to stochastic gradient Langevin dynamics? Maybe HMC type algorithms such as NUTS [4] should also be considered, since they have shown to be performing well empirically and given the number of datapoints is not too large that needs stochastic gradient .
- *Handling Variable Dimensions*: While masking is one approach to manage variable dimensions, could the authors clarify if they looked into other methods, like the ones used by [1]?
- *GMM's Variable Dimension Handling*: How does the model handle variable output dimensions in number of mixtures for GMMs, and what is the strategy for determining the number of mixtures?
- *Figure Positioning*: The positions of the figures within the paper were not specified. Could the authors provide more clarity on this aspect?
- *Comprehensive Ablation Study*: Would the authors consider conducting an ablation that sweeps on dimensionality from 1-100D?

[4] Hoffman, Matthew D., and Andrew Gelman. "The No-U-Turn sampler: adaptively setting path lengths in Hamiltonian Monte Carlo." J. Mach. Learn. Res. 15, no. 1 (2014): 1593-1623.

---

> ### Author Response · Authors · 2023-11-16
> **Author Response**
>
> We thank the reviewer for their valuable comments and hope that our response addresses and clarifies the raised concerns. We are open to further discussions to resolve any remaining comments or ambiguities.
>
> **Contribution and Metrics**
>
> We refer the reviewer to the common shared response which provides a discussion about the novelty of our work and the different contributions made as well as references to the exact formulae used to define the metrics that we consider.
>
> **Literature Gap**
>
> We thank the reviewer for bringing these related lines of work to our attention. While they share similarities in the sense that they amortize the inference process for the kernel function of Gaussian Processes (GPs), it is crucial to delineate the distinctions in our approach. Unlike these methodologies, our framework does not involve the estimation of kernel functions for GPs. Instead, we present a more comprehensive framework for Bayesian posterior estimation based on amortized inference, emphasizing the modeling of the entire posterior distribution rather than relying on point estimates. However, since our approach does share some of the underlying similarities (connections to meta learning, use of transformer architecture, etc.), we include a discussion on the mentioned approaches in Appendix A of the revised manuscript.
>
> **Variable Dimension Handling**
>
> We agree with the reviewer about predicting feature-specific parameters in modeling the posterior distribution. We had actually tried such an approach for linear regression where we leveraged attention across features as well as across observations. However, not only was it too compute intensive because of such interleaved attention operations but it also did not perform comparably to our current approach.
>
> While we do agree with the reviewer’s point that using a parameter for each feature dimension allows for leveraging the existing permutation equivariance in the feature space for a class of probabilistic models, we argue that such a setting can be fairly limited. That is, it is conceptually straightforward to learn such a system for linear regression or Gaussian Mixture models but learning such a general system that exploits all the equivariances present in more complex models (eg. Bayesian Neural Nets) is far less obvious. In particular, leveraging the methodology explored in [1] would not work for models with deep learning based probabilistic models, for which it is not clear if the parameters corresponding to some intermediate layers should be tied to certain feature dimensions, and if so, which? In particular, explicitly modeling all the equivariances present in a deep neural network is likely intractable and hence while we agree that it is a very interesting direction, we feel that it is an orthogonal line of research which would deserve a separate exploration.
>
> **GMM’s Variable Dimension Handling**
>
> We would like to clarify that although our methodology accommodates the modeling of Gaussian Mixture Model (GMM) tasks with varying feature dimensions, it does not currently support variable numbers of mixtures within the same model. We acknowledge any lack of clarity on this matter and will make necessary revisions in the draft to mitigate potential confusion.
>
> **Ablation Limitations**
>
> We would also like to clarify that for variable dimensional experiments, we choose the dimensions 1-100 uniformly so even if we were to do a sweep over these dimensions to see how the performance varies, it remains “in-distribution” testing. That being said, we take the reviewer’s point under consideration and showcase the trends over different dimensional setups in Figures 5-7 over different KL-based optimization strategies, architecture choices and variational assumptions and hope that it solves the reviewer’s concerns. For OoD testing, we refer the reviewer to the model mis-specification and tabular experiments.
>
> **Choice of MCMC**
>
> We apologize for not providing more clarity on this. We actually use Langevin Dynamics (full batch).  We also experimented with NUTS sampler but found it to be quite slow in nonlinear Bayesian Neural Network parameter estimation, especially since for each metric we would need to run the MCMC algorithm 100 times as we consider an average over 100 different datasets for each experiment.
>
> **Figure Positioning**
>
> We apologize for the figure positioning and will try to improve it where possible.
>
> [1] Liu, Sulin, Xingyuan Sun, Peter J. Ramadge, and Ryan P. Adams. "Task-agnostic amortized inference of gaussian process hyperparameters." Advances in Neural Information Processing Systems 33 (2020): 21440-21452.
>
> We hope that our response clarifies the reviewer's concerns and we would be happy to answer any additional questions and concerns.

---

> > ### Comment · Reviewer_oaRk · 2023-11-22
> >
> > Thank you for providing clarifications on GMM number of mixtures, problems with varying dimensions. I acknowledge that the paper proposes a more general framework for probabilistic models with tractable likelihoods. However, I share similar concerns with other reviewers about how this paper should be positioned in the context of literature of amortized inference. There has been much efforts of developing amortized inference framework for different models, in regression it is Neural processes and amortized GP hyperparameters (a special case of this paper's framework). It would be necessary to show how this general framework solves problems beyond previous methods. Current experiments focus on predictive performance and linear/non-linear models and GMMs, which are less interesting if the purpose of the paper's mainly focus is on empirical contributions.

---

> > > ### Author Response · Authors · 2023-11-23
> > > **Author Response**
> > >
> > > We want to thank the reviewer for the fruitful discussion and ideas mentioned. In particular, how to frame the presented novel method in the picture of other purposed methods (GP/NP) that utilize amortized inference for posterior prediction and parallel research streams like SBI that differ in the problem setup.
> > >
> > > It is correct that in a particular case, when the underlying data-generating function is a Gaussian process, and thus this framework is not reliant on a surrogate likelihood but can use the actual ground truth likelihood function, we can compare our method with GPs, especially with the focus of the presented method on amortized posterior inference.

---

### Author Response · Authors · 2023-11-16
**Common Response (1/2)**

We sincerely appreciate the reviewers’ valuable comments. In this shared response, we aim to address and clarify the majority of the common concerns raised and we are happy to engage in additional discussions should there be any remaining unresolved comments.

**Contributions**

Reviewers raised concerns regarding the novelty of our work and have sought clarification regarding the specific contributions made. We would like to take this opportunity to argue that our research identifies a notable gap in the current scientific literature relating to the amortized estimation of Bayesian posterior distributions. Existing literature predominantly engages in amortization over datasets within two overarching categories: (a) Bayesian posterior estimation through the Simulation-Based Inference (SBI) framework and (b) predictive modeling using amortized latent variable models (Neural Processes; NPs) or Gaussian Process (GP).

In contrast to the aforementioned methodologies, our approach addresses the challenge of amortized Bayesian posterior estimation from the standpoint of Variational Inference. This distinction sets it apart from the prevalent forward KL paradigm employed in SBI approaches or the maximum likelihood approach utilized in predictive modeling with NPs, where the goal is not to perform posterior estimation.

SBI is specifically tailored to address challenges inherent in scenarios where a tractable likelihood is unattainable. Conversely, NPs leverage an Expectation-Maximization (EM) style approach to optimize the parameters of the likelihood model through Maximum Likelihood Estimation (MLE). It is important to note that a significant class of probabilistic models are characterized by well-defined, computable, and differentiable likelihoods but surprisingly, there is currently no method available that effectively performs amortized Bayesian inference over the parameters of such probabilistic models. While SBI does tackle Bayesian posterior estimation, it is confined to models lacking tractable likelihoods.

In response to the identified gap in the literature concerning Bayesian posterior estimation within probabilistic models featuring tractable likelihoods, we systematically explore two distinct methodological approaches. The initial approach, denoted as Forward KL, entails the application of the Simulation-based Inference (SBI) framework to the Bayesian posterior estimation problem. We, as mentioned, introduce an alternative strategy that conceptualizes the problem as an amortized variational inference task, employing the Reverse KL methodology. Unlike NPs, our method does not involve a maximum likelihood estimation based procedure for the parameters of the probabilistic model. Instead, our primary objective is to infer the posterior distribution over them. This nuanced differentiation in methodology contributes to the unique positioning of our approach within the broader landscape of Bayesian posterior estimation methodologies.

Another contribution of our work is that we systematically perform ablations across diverse design considerations, encompassing architecture, class of distributions, and probabilistic models. While Variational Inference (VI) and amortization are established techniques in the literature, our contribution lies in their application to the specific challenge of Bayesian posterior estimation in probabilistic models with tractable likelihood functions.

Furthermore, our proposed methodology yields notable advantages in the context of model mis-specification. Beyond demonstrating enhanced out-of-distribution (OoD) generalization when assessed in a zero-shot setting, it exhibits the capability to be trained on observations for which the underlying probabilistic model is unknown. This represents a distinctive feature not present in SBI systems. For instance, in scenarios where observations stem from a linear system, but our assumption erroneously posits a nonlinear probabilistic model, SBI methods cannot be trained with this data but our proposed method can, and we show the benefits in Table 3. This capacity is particularly noteworthy in practical scenarios where we encounter observations without a priori knowledge of the underlying probabilistic model. In the Reverse KL framework, we can effectively leverage such observations by employing the assumed probabilistic model, even when it diverges from the ground-truth model.

---

> ### Author Response · Authors · 2023-11-16
> **Common Response (2/2)**
>
> Specifically, the tractability of Equation (9) hinges on the condition that the dataset $\mathcal{D}$ is sampled in accordance with the assumed probabilistic model $p(\mathcal{D} | \mathbf{\theta})$. In contrast, Equation (8) remains tractable even without this strict condition, enabling training with data from diverse sources. As is the case with most novel work, our proposed method indeed does share mathematical and experimental similarities to current existing works (eg. the concept of amortization, variational inference and transformers have been in the literature for a while now).
>
> Moreover, it is crucial to emphasize that to the best of our knowledge, none of the existing methods within SBI or NPs demonstrate the capability to effectively handle inputs of variable dimensions while our method can, which in itself is a novel contribution. Additionally, we also test for OoD generalization of such models not only through synthetic model mis-specification settings but also through their ability to transfer to real-world regression and classification problems. We believe that we are the first ones to highlight the differences in the predictive ability of Forward (SBI) and Reverse KL amortization approaches, especially under different kinds of distribution shifts.
>
> **Metrics**
>
> We thank the reviewer for bringing this up. To make things clearer and more precise, we have added Section D in the Appendix that now provides the exact metrics that we use. We would like to clarify that we use the loss and accuracy metrics on the predictive distribution and not the approximate posterior, exactly because expected $L_2$ loss in the parameter space is not suited for problems with multimodal posteriors as Reviewer iJJB pointed out. Thus, a good proxy for understanding if we capture a reasonable posterior is to see if it leads to good predictive performance. Additionally, to assess the quality of the posterior we provide some symmetric KL divergence based metrics where the true posterior is available in closed form.
>
> We hope that this addresses the reviewers' concerns regarding the novelty and contributions of this work, as well as clarifies the exact metrics that we use.

---

### Comment · Area_Chair_XP6W · 2023-11-22

Dear all,

The author-reviewer discussion period is about to end.

@authors: If not done already, please respond to the comments or questions reviewers may further have. Remain short and to the point.

@reviewers: Please read the author's responses and ask any further questions you may have. To facilitate the decision by the end of the process, please also acknowledge that you have read the responses and indicate whether you want to update your evaluation.

You can update your evaluation positively (if you are satisfied with the responses) or negatively (if you are not satisfied with the responses or share other reviewers' concerns). Please note that major changes are a reason for rejection.

You can also keep your evaluation unchanged. In this case, please indicate that you have read the responses, that you do not have any further comments and that you keep your evaluation unchanged.

Best regards,
The AC

---

### Meta-Review · Area_Chair_XP6W · 2023-12-09

**Metareview:**

The reviewers unanimously recommend rejection (5-3-3). The reviewers have concerns regarding the clarity of the presentation, the position of the paper with respect to previous work on amortized inference (in all its variants and settings), and the corresponding empirical evaluation. The author-reviewer discussion has been constructive and has led to a number of improvements to the paper. However, the reviewers remain unconvinced by the significance of the contribution. We encourage the authors to address the reviewers' comments and to take into account the suggestions made during the author-reviewer discussion period in a revised version of the paper submitted to a future conference.

**Justification For Why Not Higher Score:**

The reviewers unanimously recommend rejection (5-3-3).

**Justification For Why Not Lower Score:**

N/A

---

### Decision · Program_Chairs · 2024-01-16

Reject